# Spatial metabolomics reveals glycogen as an actionable target for pulmonary fibrosis

Lindsey R. Conroy[1,2,12], Harrison A. Clarke[3,12], Derek B. Allison[2,4,12], Samuel Santos Valenca[5,6,12], Qi Sun[1], Tara R. Hawkinson[3], Lyndsay E. A. Young[7], Juanita E. Ferreira[4], Autumn V. Hammonds[4], Jaclyn B. Dunne [8], Robert J. McDonald[4], Kimberly J. Absher [4], Brittany E. Dong[5,6], Ronald C. Bruntz[7], Kia H. Markussen [7], Jelena A. Juras[1,2], Warren J. Alilain[1,9], Jinze Liu[10], Matthew S. Gentry [2,3,7,11], Peggi M. Angel[8], Christopher M. Waters [5,6,13] ✉ & Ramon C. Sun [1,2,3,11,13] ✉

Matrix assisted laser desorption/ionization imaging has greatly improved our understanding of spatial biology, however a robust bioinformatic pipeline for data analysis is lacking. Here, we demonstrate the application of high-dimensionality reduction/spatial clustering and histopathological annotation of matrix assisted laser desorption/ionization imaging datasets to assess tissue metabolic heterogeneity in human lung diseases. Using metabolic features identified from this pipeline, we hypothesize that metabolic channeling between glycogen and N-linked glycans is a critical metabolic process favoring pulmonary fibrosis progression. To test our hypothesis, we induced pulmonary fibrosis in two different mouse models with lysosomal glycogen utilization deficiency. Both mouse models displayed blunted N-linked glycan levels and nearly 90% reduction in endpoint fibrosis when compared to WT animals. Collectively, we provide conclusive evidence that lysosomal utilization of glycogen is required for pulmonary fibrosis progression. In summary, our study provides a roadmap to leverage spatial metabolomics to understand foundational biology in pulmonary diseases.

Mapping genotype to phenotype with near single-cell resolution and spatial definition is a major innovation in next-generation life science techniques. Recent improvements in RNA sequencing technologies that allow the detailed analysis of a single isolated cell from host tissues is a major advancement in achieving this goal. The resolution of single-cell RNA sequencing (scRNAseq) technology has impressive resolution that allows interrogation of tissue complexity and interactive networks at a cellular level[1–4]. The scRNAseq workflow has

[1]Department of Neuroscience, University of Kentucky College of Medicine, Lexington, KY 40536, USA. [2]Markey Cancer Center, Lexington, KY 40536, USA. [3]Department of Biochemistry & Molecular Biology, College of Medicine, University of Florida, Gainesville, FL 32610, USA. [4]Department of Pathology and Laboratory Medicine, University of Kentucky College of Medicine, Lexington, KY 40536, USA. [5]Department of Physiology, University of Kentucky College of Medicine, Lexington, KY 40536, USA. [6]Saha Cardiovascular Research Center, University of Kentucky, Lexington, KY 40536, USA. [7]Department of Molecular and Cellular Biochemistry, University of Kentucky College of Medicine, Lexington, KY 40536, USA. [8]Department of Cell & Molecular Pharmacology & Experimental Therapeutics at the Medical University of South Carolina, Charleston, SC 29425, USA. [9]Spinal Cord and Brain Injury Research Center, Lexington, KY 40536, USA. [10]Department of Biostatistics, Massey Cancer Center, Virginia Commonwealth University, Richmond, VA 23284, USA. [11]Center for Advanced Spatial Biomolecule Research, University of Florida, Gainesville, FL 32610, USA. [12]These authors contributed equally: Lindsey R. Conroy, Harrison A. Clarke, Derek B. Allison, Samuel Santos Valenca. [13]These authors jointly supervised this work: Christopher M. Waters, Ramon C. Sun. ✉e-mail: chris.waters41@uky.edu; ramonsun@ufl.edu

allowed researchers to study complex cellular heterogeneity from diseased and healthy tissues, revealing complex and rare cell populations, while uncovering regulatory relationships and tracking distinct cell lineages during development[5–8]. Current single-cell approaches include cell-separation-based methods that are focused on molecular features such as proteins, lipids, and metabolites[9–11]. To this end, in situ-based spatial technologies are beginning to emerge that can be mapped to anatomical regions, bridging the gap between genotype to phenotype analysis[12,13].

Complex carbohydrates such as N-linked glycans and glycogen are dynamic macrometabolites that impact complex and intertwined biochemical pathways[14,15]. Anabolic pathways for the biosynthesis of glycogen and N-linked glycans span multiple cellular compartments including the nucleus, cytoplasm, endoplasmic reticulum, Golgi, and the plasma membrane[14–17]. Together, complex carbohydrates modulate a myriad of cellular functions including bioenergetics, epigenetics, protein-ligand binding, membrane transport activity, and protein turnover[18]. N-linked glycans are especially critical as structural components for anatomical regions, and are vital for immune modulation and organ function[19]. They are highly concentrated in pulmonary fibrosis and are part of the extracellular collagen, proteoglycan, and N-linked glycan matrix to support fibroblast growth and contribute to remodeled lung architecture, function, and hindering gas exchange[20,21]. Oligomerization of simple sugars to glycogen results in increased polarity and decreased solubility; therefore, quantitative, and spatial analyses of complex carbohydrates has been challenging. To address this limitation, a new technique was recently developed employing enzyme-assisted matrix-assisted laser desorption/ionization-mass spectrometry imaging (MALDI-MSI) that can simultaneously perform spatial profiling of both glycogen and N-linked glycans from formalin-fixed paraffin-embedded (FFPE) mammalian tissue sections in situ, preserving important anatomical regions[15,22].

MALDI-MSI utilizes robotic precision micromovement coupled to a Nd:YAG laser for spatial profiling of both organic and biological matter[23–25]. It is capable of high sampling aptitude with the ability to record tens of thousands of pixel-mapped spectra from a single tissue section coupled with spatial resolution[26–29]. The end result is a multidimensional, information-rich dataset containing metabolic features from normal and diseased tissues[24,30–32]. Recent advances in MALDI-MSI enable the detection of complex carbohydrates such as proteoglycans[33], N-linked glycans[34], and glycogen architecture[15,22] from FFPE tissue sections. The ability to utilize FFPE tissue sections dramatically increases the accessibility to banked clinical specimens with up to decades of patient-matched metadata. These analyses are hypothesis-generating in nature and can illuminate previously unknown regional or cellular-specific metabolic events that can be tested for translational therapeutic interventions.

In this study, we demonstrate the application of high-dimensionality reduction and spatial clustering (HDR-SC) of MALDI-MSI datasets to map histopathological regions of human FFPE specimens. Further, we identified a set of unique carbohydrate features that predict fibrotic tissue with near 99.6% accuracy in a cohort of human diseased lung tissues. Targeted analysis of enriched features in fibrotic tissues reveal unique sets of complex carbohydrates as previously unidentified metabolic hallmark of human PF tissues. Finally, using both genetically modified mouse models and the bleomycin-induced mouse model of pulmonary fibrosis (PF), we demonstrate that glycogen utilization through the lysosomal salvage pathway and metabolic channeling between glycogen and N-linked glycans are critical for fibrosis development in vivo.

## Results
### HDR-SC of MALDI-MSI datasets reveal tissue anatomical regions
Anabolic pathways for complex carbohydrates such as glycogen and N-linked glycans are critical facets of glucose metabolism, and

emerging work demonstrates that they are metabolically channeled through common substrates[14,15]. Comprehensive profiling of both can elucidate compartmentalized glucose metabolism within a cell (Fig. 1a). Recently, a technique was developed for the multiplexed imaging of N-linked glycans and glycogen using MALDI-MSI (Fig. 1b)[15,35]. This method takes advantage of two highly specific enzymes: isoamylase, which cleaves glycogen to release linear glucose chains, and peptide: N-glycosidase F (PNGaseF), which liberates N-linked glycans from proteins (Fig. S1A, B). Subsequently, linear glucose chains and N-linked glycans can be distinguished by ion mobility separation during MALDI-MSI (Fig. S1C)[15].

The multidimensional nature of MALDI-MSI datasets is highlighted by the fact that each pixel from a MALDI-MSI experiment contains a full ion spectrum of detectable molecular features (Fig. 1c–f). To further explore the full potential of the multiplexed MALDI-MSI workflow, we tested whether HDR-SC of MALDI-MSI datasets can match spatial tissue anatomy using MALDI-MSI of a human liver section (Fig. 1g). First, a carefully curated and established list of 50 MALDI matrix peaks (m/z) and 155 glycogen and N-linked glycan peaks (m/z) were selected for HDR-SC analysis (Table S1A, B), then we performed peak integration to account for mass drift during the MALDI scan and improve image resolution (Fig. S1D and E). Normalized input data (Fig. 1h) undergoes dimensionality reduction and clustering using the Leiden clustering algorithm[36], and the results are presented as a spatial plot and a Uniform Manifold Approximation and Projection (UMAP) plot (Fig. 1i)[37]. UMAP visualization can accurately identify α-cyano-4-hydroxycinnamic acid (CHCA) matrix clusters (often presented as a standalone cluster on the UMAP plot) and can be omitted from further analyses (Fig. 1j). Spatial extrapolation of remaining clusters based on pixel coordinates revealed unique spatial regions within the liver section (Fig. 1k).

To test whether unique pixel clusters match known liver anatomical regions and structures, we performed detailed histology analysis using a liver section stained with hematoxylin and eosin (H&E) followed by annotation by expert clinical surgical pathologists (Fig. S1F, G). By matching to histology evaluation in a blinded fashion, clustering results matched accurately with histological features such as liver lobules, smooth muscles, and portal veins (Fig. S1H, I). Interestingly, unique clusters were observed for inner and outer layers within the liver lobules (cluster 6 and 1, respectively) and separated both longitudinal and crosscut sections of the portal vein surrounded by smooth muscle (cluster 12 and 0, respectively) (Fig. S1J). Glycogen features are predominately enriched in the liver lobule clusters while smooth muscle and portal veins are enriched for N-linked glycans (Fig. S1K). Finally, we performed glycogen structural analyses between inner and outer liver lobules and portal veins. As expected, the hepatocytes within liver lobules have both shorter and longer chain length distribution than at the region near the portal vein. The hepatocytes within the central portion of the lobule have higher short glycogen chain abundance than the outer layer; however, there were no differences between longitudinal and cross-section portal vein regions (Fig. S1L). These findings demonstrate the ability of HDR-SC of MALDI-MSI datasets to accurately identify tissue microanatomy.

### HDR-SC reveals major histopathology of human lung diseases
To test whether HDR-SC analysis can identify pathological changes in human diseased tissue, we performed MALDI-MSI on lung tissue sections from multiple human idiopathic pulmonary fibrosis (IPF) patients and a single COVID-19 patient (Figs. 2a, f and S2A, D). For each human tissue section, HDR-SC identified different clusters between the IPF (Figs. 2b and S2B) and COVID-19 (Figs. 2g and S2E) patient samples. These clusters matched accurately to smooth muscle (cluster 0), end-stage fibrosis (cluster 5), diffuse alveolar damage (DAD) (cluster 6), mucin aggregates in the airway (cluster 10), and edematous loose stroma (cluster 11) (Fig. 2c) in the IPF patient tissue. In addition,

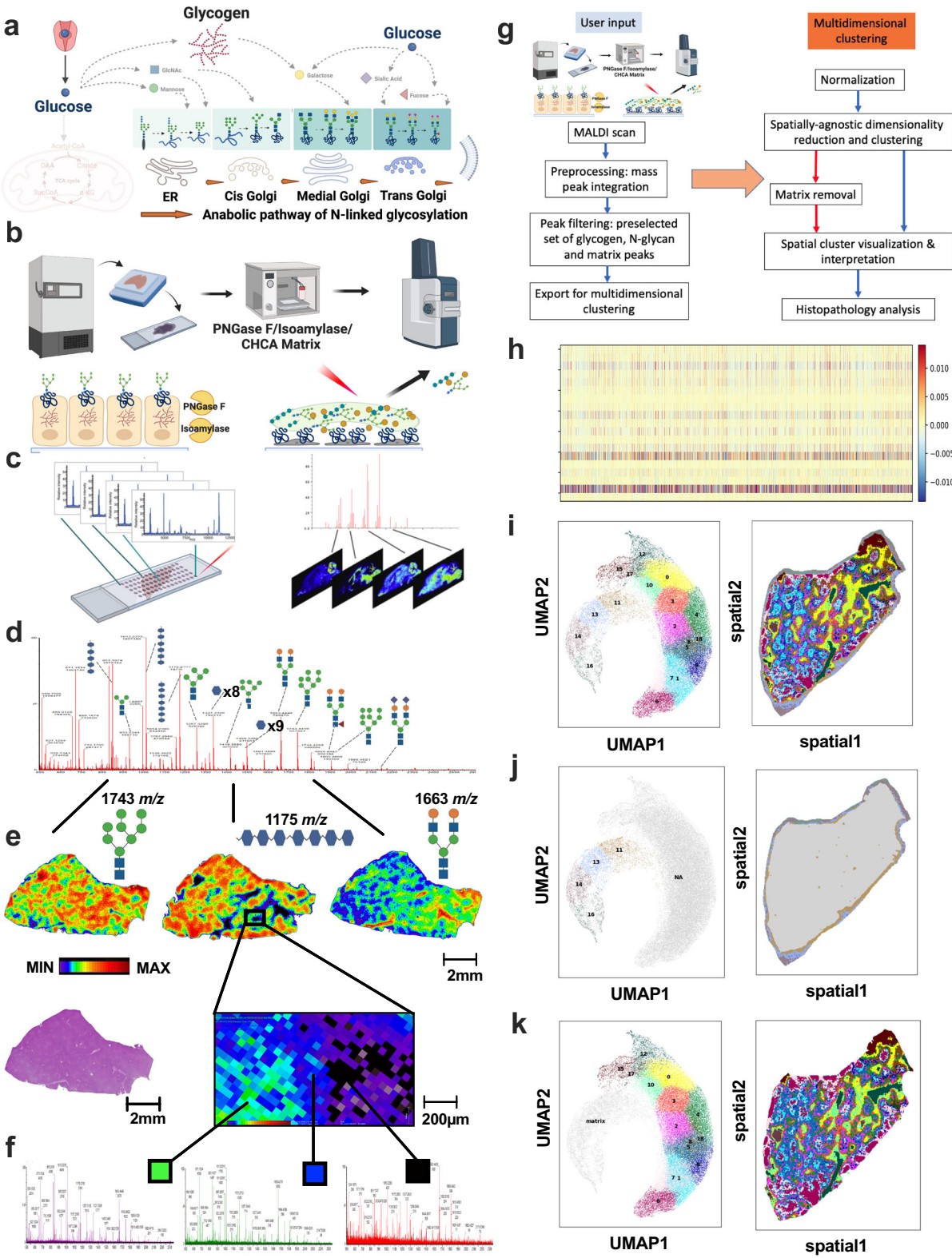

clusters matched accurately to early (cluster 1), mid (cluster 0), and end-stage fibrosis (clusters 3 and 4) as well as DAD (clusters 6 and 8) and mucin aggregates in the airway (cluster 12) in the COVID-19 patient (Fig. 2h). Notably, HDR-SC analysis accurately identified shared pathologies between IPF and COVID-19 tissues including DAD, end-stage fibrosis, and mucin aggregation (Figs. 2d, e and 2i, j). Histopathology analysis of additional IPF and COVID-19 tissues following HDR-SC, summarized in Fig. S2C and F, also accurately identified end-

stage fibrosis as a major histopathological feature of both lung diseases based on unique complex carbohydrate features (Fig. S2G–J).

## Glycogen as a clinical hallmark of PF from different disease origins

Pulmonary fibrosis (PF) is a severe and terminal disease that currently lacks a cure. There is a critical need to identify therapeutic vulnerabilities within PF for the future development of small molecule

**Fig. 1 | MALDI imaging of complex carbohydrates is a multidimensional dataset. a** Metabolic pathways of complex carbohydrate metabolism inside a cell. Anabolic metabolism of glycogen and N-linked glycans through the endoplasmic reticulum (ER) and Golgi are critical for a multitude of cellular functions. Created with BioRender.com. **b** Schematic of the workflow for multiplexed imaging of glycogen and N-glycans using formalin-fixed paraffin-embedded (FFPE) specimens. Tissues are sectioned onto a microscope slide and are co-treated with peptide: N-glycosidase F (PNGase F) to cleave and release N-glycans and isoamylase to cleave α-1,6-glycosidic bonds releasing linear oligosaccharide chains. Following application of α-cyano-4-hydroxycinnamic acid (CHCA) ionization matrix, samples are analyzed by MALDI and quadrupole time-of-flight mass spectrometry. Created with BioRender.com. **c** Left: Schematic of molecular ions recorded in each pixel after laser desorption ionization. Right: Spatial heatmap of a specific biological feature (*m/z*) extrapolated from total ion current (TIC) after MALDI-MSI of tissue sections. Created with BioRender.com. **d** TIC representing the sum of all pixels after MALDI-MSI of a human liver section. **e** Spatial heatmap images of selected complex carbohydrate ions (*m/z*) corresponding to either glycogen or N-linked glycans selected from (**d**). *m/z* values and molecular structures of selected oligosaccharide chain and N-linked glycans are on top of the heatmap. An adjacent liver section stained with hematoxylin and eosin (H&E) is presented below the heatmap for 1743 *m/z*. Scale bar: 2 mm. **f** TIC extracted from three unique pixels based on the spatial heatmap presented in (**e**). **g** Simplified flow chart for the high-dimensionality reduction and spatial clustering (HDR-SC) analysis workflow. The user input module includes MALDI-MSI and data curation, followed by the computer-based module that utilizes unsupervised clustering and spatial annotation of unique clusters. Created with BioRender.com. **h** Representative input data for HDR-SC presented as heatmap to show heterogeneous complex carbohydrate abundance. 5% of total pixels (columns) and mixture of matrix and carbohydrate *m/z* (rows) were shown as heatmap for ease of visualization. **i** UMAP and spatial plots of the human liver specimen by HDR-SC analysis. **j** UMAP and spatial plots showing matrix only cluster from the human liver specimen by HDR-SC analysis. **k** UMAP and spatial plots after matrix removal.

inhibitors. Based on HDR-SC, we performed metabolite enrichment analysis among each pixel within different annotated clusters and examined overrepresented carbohydrate features specific to fibrotic regions (Fig. 2k). We identified a number of core fucosylated N-linked glycans upregulated among fibrotic regions between different patients (Figs. 2k and S2K, L). This finding supports previously published results that fucosylation is involved during fibrosis disease progression[38–41]. Interestingly, we observed a number of glucose polymers appear in the overrepresented metabolite list which correspond to glycogen accumulation in the fibrotic region (Fig. 2k, l). Next, we further evaluated this spatial glycogen phenotype in additional IPF (*n* = 3) and COVID-19 (*n* = 1) tissues and confirmed that glycogen-rich regions correspond with fibrotic regions from adjacent H&E sections annotated by a board-certified clinical pathologist (Fig. 2m, n). To confirm that glycogen is uniquely accumulated in the myofibroblast, we performed co-localization analysis with immunofluorescence (IF) using an anti-glycogen antibody[15,42] and an anti-alpha-smooth muscle actin (α-SMA) antibody[43]. We observed robust co-localization between glycogen and α-SMA (Figs. 2o and S2M), which supports the notion that glycogen is uniquely present in the myofibroblast cellular population within pulmonary fibrosis. These findings raise the interesting hypothesis that glycogen is a critical hallmark metabolite in PF and its role in PF should be further evaluated. To establish the absolute amount of glycogen in fibrotic and non-fibrotic regions of human and mouse specimens, we directly spotted increasing concentrations of purified liver glycogen as quantification standards adjacent to tissue sections followed by MALDI imaging (Fig. S2N). Using this approach, we demonstrates the dynamic linear range for glycogen by MALDI imaging from 1 ng to 1000 ng (Fig. S2N). Finally, we determined the absolute glycogen levels to be ~1200 ng/pixel in fibrotic regions and ~100 ng/pixel in non-fibrotic regions within human specimens (Fig. S2N).

To further establish that glycogen is a metabolic hallmark of lung fibrosis, we purchased a commercial tissue microarray (TMA) comprised of both human fibrotic tissue cores (*n* = 26) and normal adjacent lung resected at surgery (*n* = 7) (Fig. S3A–C). We performed MALDI-MSI on the cohort of human specimens from the TMA and performed glycogen and N-linked glycan analyses. In agreement with IPF/COVID-19 larger tissue analyses, we observed on average a 2.4-fold increase in glycogen within the fibrotic cores compared to the non-fibrotic normal lung controls (Fig. 3a, b). In addition, glycogen structure analysis shows significant increases across the spectrum of glycogen chain lengths in fibrotic cores compared to normal lung tissue (Fig. 3c). Both partial least squares-discriminant analysis (PLS-DA) and unsupervised clustering heatmap analyses using glycogen-derived glucose polymers and N-linked glycans exhibit clear separation between fibrosis and normal samples (Figs. 3d, e, S3D). Further, we performed receiver operating characteristic (ROC) analysis to assess the clinical diagnostic ability of glycogen and N-linked glycans as predictive features of lung fibrosis. While glycogen alone shows an exceptional predictive AUC value of 0.94 (Fig. S3E), glycogen in combination with N-linked glycans shows an even better predictive AUC value of 0.996 (Fig. 3f). Finally, we also confirmed that core fucosylated N-linked glycans were upregulated in fibrotic tissue cores from the TMA similar to previous results in our IPF/COVID-19 patient samples (Figs. 3g–j, S3F). Collectively, our data support the notion that glycogen-associated N-linked glycans are clinical hallmarks of pulmonary fibrosis that warrant further studies into its therapeutic potentials.

## Bleomycin-induced pulmonary fibrosis display increased glycogen

Based on human specimens, we identified extensive changes in N-linked glycan and glycogen accumulation in PF (Fig. 3b–j). We have previously established that glycogen and N-linked glycans are metabolically channeled and glycogenolysis or glycogen utilization provides substrates for N-linked glycosylation[15,35]. Therefore, we hypothesize that glycogen and N-linked glycan phenotypes observed in PF are linked and a critical step during the progression of fibrosis, and by preventing glycogen utilization we could inhibit fibrosis development, and, in theory, glycogen could be a potential therapeutic target for the treatment of fibrosis. To test this hypothesis, we utilized a common preclinical animal model of fibrosis resulting from intratracheal administration of bleomycin[44,45]. Administration of bleomycin induces an initial dose-dependent acute lung injury in the first seven days, followed by the development of fibrosis over the next 14 days[45]. First, we assessed whether mice with PF exhibit similar complex carbohydrate features compared to human fibrosis. Mice were intratracheally administered either bleomycin (0.3 U/kg) or saline and sacrificed 24 h after day 21, followed by surgical removal of the lungs (Fig. 4a). Mice were weighed daily and monitored for signs of impaired mobility, reduction in size/body weight, difficulty breathing, and/or generalized cachexia. Bleomycin treatment generated the predicted fibrotic response by day 21, determined by H&E staining and pathological assessment (Figs. 4b and S4A, B). Initial multivariate analyses assessed by MALDI-MSI demonstrate clear separation between saline and bleomycin-treated mouse lungs by both PLS-DA and unsupervised clustering heatmap analysis (Fig. S4C–E). Similar to human specimens, the bleomycin PF model displays increased core fucosylated N-linked glycans (Fig. 4c, d), and, more, importantly, the bleomycin PF model exhibits a 2.4-fold increase in glycogen stores (Fig. 4e–g), similar to those identified in human specimens (Fig. 3g–j). Also similar to human specimens, co-localization between glycogen and α-SMA was observed in the mouse model of PF (Fig. 4h), suggesting glycogen is uniquely localized with myofibroblast cells. Finally, we performed targeted liquid-chromatography mass spectrometry analysis using pooled PBS and bleomycin-treated mouse lungs (Fig. S4F). Metabolites associated

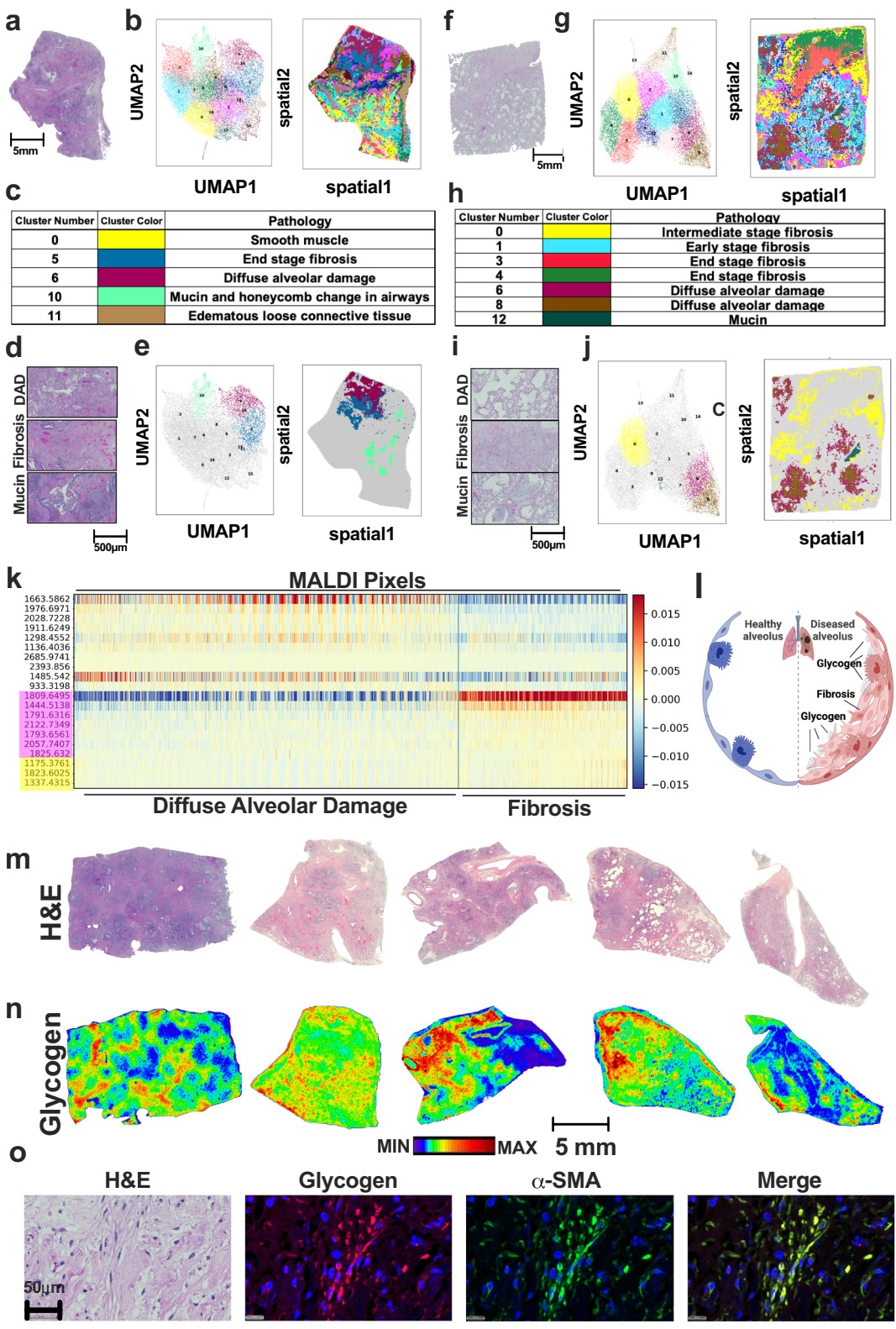

with glycogen metabolic pathways such as UDP-glucose and glucose-6-phosphate were significantly decreased at the onset of fibrosis in bleomycin-treated mice (Fig. S4F). Interestingly, amino acids such as alanine, glutamic acid, and glutamine remain unchanged between the two groups (Fig. S4F). Collectively, our data suggest similarities in complex carbohydrate metabolism between the PF model and human disease (Fig. 4k), and thus justify the bleomycin-induced PF preclinical model to study glycogen and N-linked glycan metabolism in PF.

**Inability to utilize glycogen blunts fibrosis development in vivo**
To test whether glycogen utilization is necessary for fibrosis disease progression, we employed a mouse model of glycogen storage disease lacking the glycogen phosphatase laforin ($Epm2a^{-/-}$). Glycogen dephosphorylation is required for efficient glycogenolysis and the release of glucose-6-phosphate by glycogen phosphorylase (GP)[46,47]. Laforin deficiency results in hyperphosphorylated glycogen aggregates that are inaccessible to glycogen degradation enzymes[15,48–50].

**Fig. 2 | High-dimensionality reduction and spatial clustering (HDR-SC) analysis of tissue sections from idiopathic pulmonary fibrosis (IPF) and COVID-19 patients. a** Hematoxylin and eosin (H&E) staining of an immediate adjacent IPF section used for MALDI-MSI for histopathology assessment. Scale bar: 5 mm. **b** All identified clusters visualized by UMAP (left) and spatial plots (right) in an immediate adjacent IPF section. **c** Annotation of spatial clusters to histopathology by a panel of board-certified pathologists in an immediate adjacent IPF section. **d** Zoomed in images of H&E staining of IPF section showing diffuse alveolar damage (DAD), end-stage fibrosis, and mucin aggregates in an immediate adjacent IPF section. Scale bar: 500 μm. **e** UMAP (left) and spatial (right) plots highlighting DAD, end-stage fibrosis, and mucin aggregate clusters. **f** H&E staining of an immediate adjacent COVID-19 section used for MALDI-MSI for histopathology assessment. Scale bar: 5 mm. **g** All identified clusters visualized by UMAP (left) and spatial plots (right) in an immediate adjacent COVID-19 section. **h** Annotation of spatial clusters to histopathology by a panel of board-certified pathologists in an immediate adjacent COVID-19 section. **i** Zoomed in images of H&E staining of COVID-19 section showing DAD, end-stage fibrosis, and mucin aggregates in an immediate adjacent COVID-19 section. Scale bar: 500 μm. **j** UMAP (left) and spatial (right) plots highlighting DAD, end-stage fibrosis, and mucin aggregate clusters. **k** Differentially expressed features among HDR clusters, columns represent pixels within each cluster and row is the glycogen or N-glycan feature overrepresented. Glycogen (yellow) and core fucosylated N-linked glycan (pink) features were overrepresented in fibrotic clusters. **l** Schematic of glycogen accumulation in pulmonary fibrosis. Created with BioRender.com. **m** H&E staining of additional IPF (*n* = 3, left) and COVID-19 (*n* = 2) tissue sections used for MALDI-MSI for histopathology assessment. Scale bar is below in (**n**). **n** Spatial distribution and heatmap of glycogen chain length +7 (1175 *m/z*) in additional patient tissues shown in (**m**). Scale bar: 5 mm. **o** Immunofluorescent/co-localization analysis of glycogen and alpha smooth muscle actin (α-SMA) from an adjacent 20 μm section of IPF specimen shown in (**a**). Tissue is stained with glycogen (red), α-SMA (green), and DAPI (blue).

Thus, the mice accumulate glycogen that they are unable to degrade. We administered bleomycin to both WT and *Epm2a*$^{-/-}$ mice (LKO) and collected lungs 24 h after day 21 post-treatment similar to the previous experiment (Fig. 5a). During the experiment, bleomycin-treated LKO mice did not lose as much weight compared to the bleomycin-treated WT mice (Fig. 5b). Further, three WT mice were euthanized during the treatment window due to >20% weight loss, while none of the LKO mice met the criteria for euthanasia (Fig. 5c). Following euthanasia and lung dissection, we performed H&E histopathological staining and MALDI-MSI on the lungs of all four cohorts of mice (Fig. S5A, B). Next, we performed targeted analysis in only the fibrotic regions between both WT and LKO mice guided by H&E image of an adjacent lung section. As predicted, we observed a significant buildup of glycogen stores in the LKO bleomycin-treated cohort, due to the inability to utilize glycogen in this mouse model (Fig. 5d–f). Increased glycogen within the LKO bleomycin cohort correlated with a 2-fold reduction in fibrosis burden when compared to WT mice treated with bleomycin based on histopathological assessments and Ashcroft scoring (Fig. 5g, h).

Fibrosis progression relies on remodeling of the lung extracellular matrix proteins[51]. Primarily, extensive collagen deposition within the alveolar airways provides the necessary structural and mechanical support for the chronic proliferation of diseased fibroblasts[52]. While mechanisms behind increased collagen synthesis during PF remain to be elucidated, it is known that the degree of collagen lattice correlates with disease severity[53]. In our study, we performed total collagen quantitation from the bronchoalveolar lavage fluid (BALF) and observed similar trends between LKO and WT bleomycin-treated cohorts (Fig. 5i). To confirm that the reduction of collagen is from the fibrotic regions, we performed spatial proteomics imaging using collagenase digestion to specifically interrogate collagen isoforms within the fibrotic regions[54,55] (Fig. 5j). Similar to established biology, we found significant enrichment of collagen peptides within the fibrotic regions of the lung that correlated with the Ashcroft score (Fig. S5B). Interestingly, LKO animals treated with bleomycin have much lower levels of collagen content within the fibrotic regions compared to WT bleomycin cohort when normalized to number of pixels within the fibrotic region (Fig. S5B). Further, we saw significant decreases in peptides derived from collagen 1A1/2 A (Fig. 5k), 3A1 (Fig. 5k), 5A2 (Fig. S5C), suggesting blocking glycogen utilization results in the down-regulation of more than one type of collagen. This phenotype is further validated by principal component analysis (PCA) and unsupervised clustering heatmap analyses, showing global reduction of collagen peptides within the fibrotic regions of LKO mice (Fig. S5D, E). Collectively, multidimensional spatial metabolomics and proteomics analyses suggest collagen deposition during PF is linked to glycogen metabolism and an orthogonal method to confirm the lack of endpoint fibrosis in LKO mice.

## Acid alpha-glucosidase (GAA) drives glycogen utilization in PF

Elucidating the route of glycogen utilization is needed to improve our understanding of PF biology and future translational opportunities. First, we performed immunohistochemical staining to assess the levels of glycogen synthase (GYS) and glycogen phosphorylase muscle and brain isoform (PYGM and PYGB, respectively) at the protein level[17]. We observed increased GYS protein expression within the fibrotic regions when compared with the non-diseased regions from the same tissue section (Fig. S6A, B). However, we did not observe similar increases for PYGM and PYGB at the protein level (Fig. S6A, B). To further assess glycogen metabolism in disease fibroblasts, we performed both total pool and $^{13}C$-glucose traced glycogen in both normal fibroblasts and disease fibroblasts isolated from normal and IPF patients, respectively, using previously established mass spectrometry methods[56,57]. Similar to patient specimens, diseased fibroblasts display a glycogen-rich phenotype (Fig. S6C). We then performed $^{13}C$-glucose to glycogen enrichment followed by a washout analyses that represents the rate of glycogen biosynthesis and degradation, respectively. Diseased fibroblasts display significantly higher enrichment and washout rates compared to the normal fibroblast (Fig. S6D–F), suggesting increased metabolic demand for glycogen in diseased fibroblasts with increased glycogen biosynthesis and degradation. To further support glycogen is important during fibrosis progression, we identified regions of early, mid, and end-stage fibrosis from our existing patient tissue sections (Fig. S6G). In agreement with our hypothesis, there is a stepwise decrease in glycogen stores from early to end-stage fibrosis (Fig. S6H). In contrast, we observed stepwise increases in biantennary N-linked glycans in tissue regions progressing from early to end-stage fibrosis (Fig. S6H). Collectively, stage-dependent decreases in glycogen patient tissue sections offers additional evidence to support our hypothesis that glycogen is utilized during fibrosis disease progression.

Increased rate of glycogen utilization without increases in GP protein expression, a key glycogen degradation enzyme, suggested an alternative route of glycogen degradation in the diseased fibroblasts. In our IF analysis using the anti-glycogen antibody, we observed round/puncta glycogen within the cytoplasm of myofibroblasts (Figs. 6a, b and S7B). This pattern has been described during glycogen phase separation[58] and its targeting into lysosomes[59,60]. Since GAA is the lysosomal enzyme responsible for glycogen degradation (Fig. 6c)[61,62], we performed co-localization analysis of GAA and glycogen using IF in human patient specimens. Extensive co-localization between glycogen and GAA suggests lysosomes are the site of glycogen utilization in PF (Fig. 6a). To confirm this phenotype in mice, we performed the same co-localization analysis using the lysosomal marker LAMP2. Similar to human results, mouse PF exhibited extensive co-localization between glycogen and LAMP2 (Figs. 6b and S7C), further supporting that glycogen is utilized in the lysosome in both human and mouse PF.

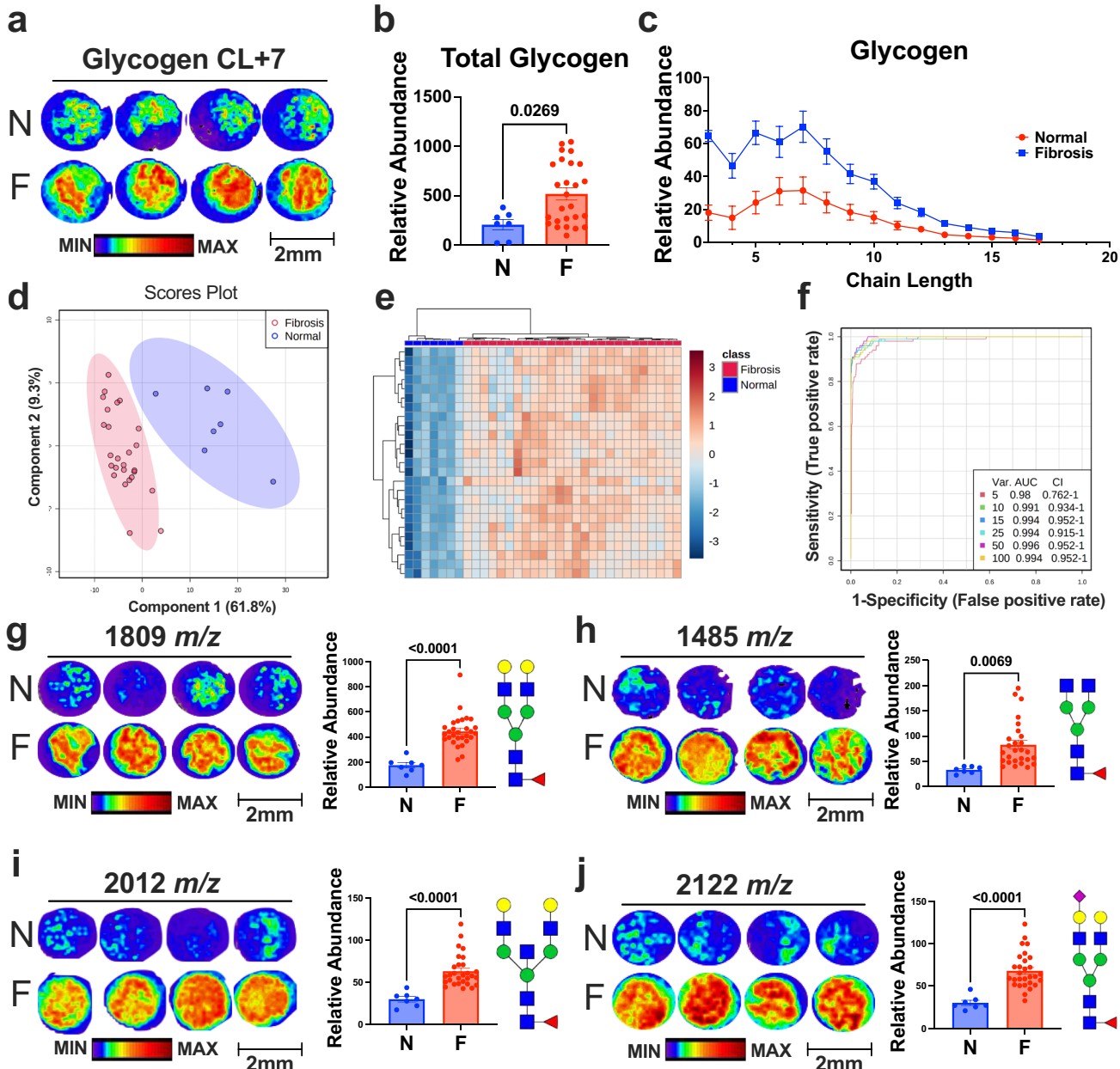

**Fig. 3 | Pulmonary fibrosis patient tissue exhibits aberrant complex carbohydrate features. a** Representative spatial distribution and heatmap of glycogen chain length +7 (1175 *m/z*) in normal (N) and fibrotic (F) lung patient tissue cores from TMA. Scale bar: 2 mm. **b** Total glycogen abundance in normal and fibrotic lung patient tissue measured by MALDI (Normal: *n* = 7, Fibrosis: *n* = 26). Values are presented as mean +/− standard error. *p*-value was calculated using two-tailed *t*-test. **c** Glycogen chain length abundance in normal and fibrotic lung patient tissue (Normal: *n* = 7, Fibrosis: *n* = 26). Values are presented as mean +/− standard error. **d** Multivariate analysis of glycogen and N-linked glycan features in normal and fibrosis lung patient samples by partial least squares-discriminant analysis (PLS-DA) displaying 95% confidence intervals. **e** Unsupervised clustering heatmap analysis of the top 25 glycogen and N-linked glycan features in normal and fibrotic lung patient. **f** Multivariate receiver operating characteristic (ROC) curve of all glycogen and N-linked glycan features between normal and fibrosis patients. **g** (Left)

Representative spatial distribution and heatmap of 1809 *m/z* in N and F lung patient tissue. (Right) Relative abundance of 1809 *m/z* in N and F lung patient tissue. Molecular structure of the selected N-linked glycan is to the right of the graph. **h** (Left) Representative spatial distribution and heatmap of 1485 *m/z* in N and F lung patient tissue. (Right) Relative abundance of 1485 *m/z* in N and F lung patient tissue. Molecular structure of the selected N-linked glycan is to the right. **i** (Left) Representative spatial distribution and heatmap of 2012 *m/z* in N and F lung patient tissue. (Right) Relative abundance of 2012 *m/z* in N and F lung patient tissue. Molecular structure of the selected N-linked glycan is to the right. **j** (Left) Representative spatial distribution and heatmap of 2122 *m/z* in N and F lung patient tissue. Scale bar: 2 mm. (Right) Relative abundance of 2122 *m/z* in N and F lung patient tissue. Molecular structure of the selected N-linked glycan is to the right. **g–j** Values are presented as mean +/− standard error (Normal: *n* = 7, Fibrosis: *n* = 26). *p*-value was calculated using two-tailed *t*-test.

The lysosome participates in the intracellular salvage pathways providing substrates for complex carbohydrate production (Fig. 6c). Given this known channeling, we hypothesized that glycogen utilization by GAA in the lysosomal salvage pathway may be necessary for fibrosis disease progression. Therefore, we employed a mouse model lacking

GAA (*Gaa*[−/−]). We administered bleomycin to both WT and *Gaa*[−/−] mice (GKO) and collected lungs 24 hours after 21 days post-treatment similar to the previous experiment (Fig. 6d). Following euthanasia and lung dissection, we performed H&E histopathological staining and MALDI-MSI on the lungs of all four cohorts of mice (Fig. S7A). Strikingly, GKO

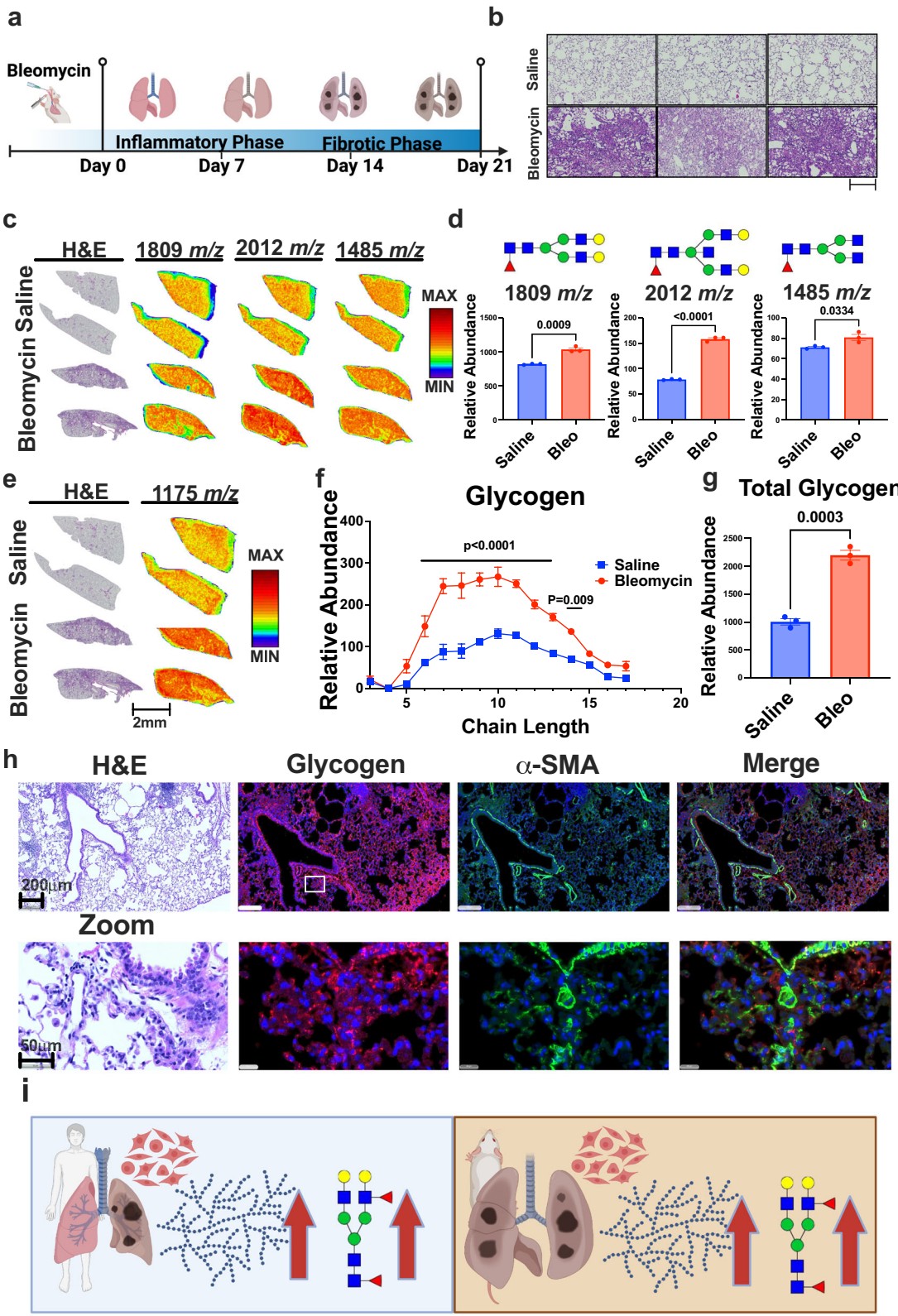

mice exhibited nearly 5-fold reduction in fibrosis burden when compared to WT mice treated with bleomycin based on histopathological assessments and Ashcroft scoring (Fig. 6e, f and S7D). Interestingly, we observed high basal level glycogen in the GKO saline-treated cohort, but glycogen did not further increase in the bleomycin-treated cohort, possibly due to the lack of endpoint fibrosis formation (Fig. 6h).

To confirm that the lysosomal salvage pathway is compromised in GKO and LKO mice, we performed MALDI MSI to assess the N-linked

glycome of the lung in both GKO and LKO cohorts of bleomycin and saline-treated animals. We observed board spectrum reduction in a wide variety of N-linked glycans in both GKO and LKO bleomycin-treated lungs when compared to the WT bleomycin-treated cohort (Fig. S6E), and the effect size was much greater in the GKO cohort. For example, N-linked glycans 1444 and 2012 m/z did not show increases in bleomycin-treated GKO mice compared to GKO saline-treated mice (Fig. 6i) but trending slightly higher in the LKO cohorts within the same

**Fig. 4 | Bleomycin-induced lung fibrosis shares same complex carbohydrate features as human fibrosis. a** Schematic of bleomycin-induced lung fibrosis model. Created with BioRender.com. **b** Zoomed in images of hematoxylin and eosin (H&E) staining of lung tissue from control (saline) and bleomycin-treated mice. Scale bar: 200 μm. **c** (Left) H&E staining of whole lung tissue from control (saline) (top) and bleomycin-treated (bottom) mice. (Right) Spatial distribution and heatmap of N-linked glycans: 1809, 2012, and 1485 *m/z* in lung tissue from control (saline) (top) and bleomycin-treated (bottom) mice. Scale bar: 2 mm. **d** Total abundance of N-linked glycans: 1809, 2012, and 1485 *m/z* in lung tissue from control (saline) and bleomycin-treated mice. Molecular structure of the selected N-linked glycans are above their respective graphs (*n* = 3 animals/group). Values are presented as mean +/− standard error. *p*-value was calculated using two-tailed *t*-test. **e** (Left) H&E staining of whole lung tissue from control (saline) (top) and bleomycin-treated (bottom) mice. (Right) Spatial distribution and heatmap of glycogen chain length +7 (1175 *m/z*) in lung tissue from control (saline) (top) and bleomycin-treated

(bottom) mice. Scale bar: 2 mm. **f** Glycogen chain length abundance in lung tissue from control (saline) and bleomycin-treated mice. (*n* = 3 animals/group). Values are presented as mean +/− standard error. *p*-value was calculated using two-way ANOVA following by multiple comparisons testing. **g** Total glycogen abundance in lung tissue from control (saline) and bleomycin-treated mice. Values are presented as mean +/− standard error. *p*-value was calculated using two-tailed *t*-test. **h** Immunofluorescent/co-localization analysis of glycogen and alpha smooth muscle actin (α-SMA) from an adjacent 20; μm section of PF mouse lung shown in **c**. Tissue is stained with glycogen (red), α-SMA (green), and DAPI (blue) following by whole slide scanning and visualized using the HALO software. Zoomed in view shown below, and the white box in the glycogen panel represents the field of view magnified. Scale bar: 200 μm and 50 μm, respectively. **i** Schematics of shared glycogen and N-linked glycogen phenotype between mouse and human PF. Created with BioRender.com.

comparison (although *P* value is greater than 0.05). Collectively, these preclinical data from two animal models of pulmonary fibrosis support the hypothesis that glycogen utilization through the lysosomal salvage pathway is critical for fibrosis development (Fig. 6k) and in agreement with the metabolic phenotype identified in human PF specimens.

## Discussion

In this study, we applied multidimensional spatial metabolomics analyses to demonstrate application for histopathology prediction and biomarker identification in human and mouse diseased tissues. We chose spatial metabolomics analysis of complex carbohydrates for a number of reasons: (1) The ability to utilize stored FFPE tissue samples significantly increases accessibility to a variety of human disease clinical specimens[34,63]. (2) Complex carbohydrate metabolism is intimately connected to glucose utilization and spans multiple cellular compartments[14]. (3) Complex carbohydrates are critical structural metabolites for cell-cell adhesion, signaling, and extracellular matrix remodeling[14,64]. (4) Aberrant complex carbohydrate metabolism drives pathogenesis in multiple human diseases including neurodegeneration[65,66], type II diabetes[67], cardiovascular disease[68], and several types of cancer[69,70]. In this study, we demonstrate HDR-SC and biomarker prediction using spatial metabolomics datasets. HDR-SC accurately captured fibrotic regions in multiple tissue sections analyzed when matched to histopathology annotation. Further, differential feature analysis identified biomarkers that highlight a potential role for glycogen utilization during fibrosis disease progression. Using two separate mouse model of fibrosis, we demonstrate that the inability to utilize glycogen leads to significantly blunted N-linked glycan and collagen content and fibrosis development in vivo.

MALDI-MSI is a high-dimensional dataset, similar to the scRNAseq format[71] wherein complex carbohydrates are biological features stored within each pixel, similar to gene expression levels within each cell. However, there are several important differences. First, each pixel from MALDI-MSI is recorded with precise X-Y coordinates to allow true spatial mapping of anatomical regions. Second, multiplexed glycogen and N-linked glycan MALDI-MSI takes advantage of a large collection of well-annotated human clinical specimens with decades of clinical metadata in the form of FFPE tissue. The ability to utilize FFPE tissue also bypasses the need to collect fresh tissue from surgery for human clinical studies. The in situ nature of MALDI-MSI also avoids lengthy and potentially destructive cell isolation steps, thus preserving anatomical regions. Of note, many histopathological features such as end-stage fibrosis and mucin patches contain large fractions of non-cellular features. For example, the presence of mucin within airways and in regions of honeycomb change contain abundant glycoproteins. These features are often lost in single-cell analyses but are preserved by the MALDI-MSI workflow. To this end, what MALDI-MSI lacks in cellular

information could be complimented with scRNAseq, multiplexed ion beam imaging by time of flight (MIBI-TOF), or co-detection by indexing (CODEX) analyses[72]. Co-analysis or vertical integration with single-cell techniques would enhance cellular metabolic origins that are related to the etiologies of diseases.

Pulmonary fibrosis (PF) is the long-term consequence of genetic mutations or acute damage to the lung from environmental exposures, including radiation/chemotherapy[73,74], acute respiratory distress syndrome (ARDS)[75], cancer[76], and, more recently, severe lung disease caused by the SARS CoV-2 virus (COVID-19)[77]. In some cases, PF arises from undefined causes and is classified as IPF[78]. The pathology of PF includes remodeling of the alveolar regions with excessive matrix production such as N-glycans and collagen by fibroblasts, myofibroblasts, and mucin aggregation that lead to severe thickening of the alveolar septa of the lung[79,80]. PF patients suffer from poor gas exchange and subsequently low tissue oxygenation that leads to other co-morbidities and death[81]. Our pipeline identified unique subclasses of complex carbohydrates enriched in myofibroblasts of PF patient samples. In our analyses, core fucosylated N-glycans and glucose polymers from glycogen are differentially enriched in the myofibroblasts of PF. Using a separate TMA containing fibrosis and normal tissues, we demonstrate exceptional binary predictability of N-glycans and glycogen for the diagnosis of fibrosis. Future studies should focus on the predictability in multivariant pathologies such as acute fibrinous and organizing pneumonia (AFOP), DAD, and mucinous lesions. We anticipate that artificial intelligence (AI) would aid in the development process of implementing spatial metabolomics to digital pathology.

Current PF treatment relies on steroids that only temporarily delay fibrosis progression[82], and there are no effective treatments to reverse the clinical course of the disease, resulting in permanent loss of lung function[78]. MALDI-MSI analyses of human pulmonary disease specimens revealed a glycogen-rich phenotype within the fibrotic regions. Interestingly, mouse models of PF through bleomycin-induction also displayed a similar glycogen-rich phenotype. Through a series of molecular, cellular, and animal modeling experiments, we concluded that there is an increased metabolic demand for glycogen metabolism in diseased fibroblasts and highlighted that glycogen utilization by GAA through the lysosomal pathway is a critical metabolic process that occurs during pulmonary fibrosis. Mice deficient in GAA did not form PF after bleomycin treatment. The lysosomal salvage pathway provides substrates for the synthesis of proteoglycans[83] and N-linked glycans[14] that are critical components of the extracellular matrix. Coincidentally, myofibroblasts drive extracellular remodeling of PF[43], and increased glycogen utilization through the lysosomal salvage pathway would support the metabolic demand for extracellular matrix remodeling. This hypothesis is supported by the result demonstrating that Gaa⁻/⁻ mice blunted the increase in N-linked glycans after bleomycin treatment. We propose that this phenotype is

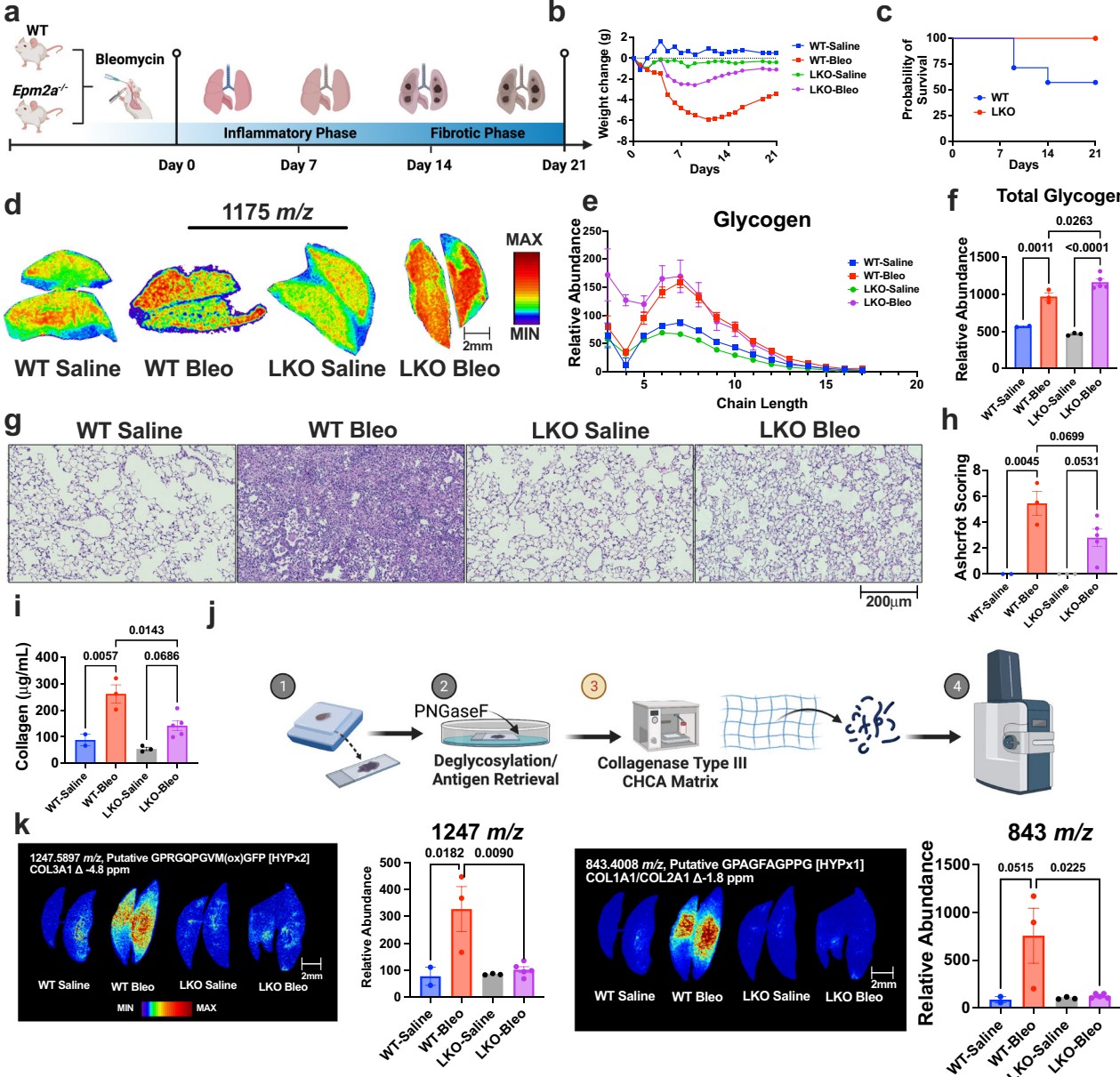

**Fig. 5 | Loss of laforin blunts complex carbohydrate perturbations during bleomycin-induced lung injury. a** bleomycin-induced lung fibrosis model in wild-type (WT) and *Epm2a⁻/⁻* (LKO) mice. Created with BioRender.com. **b** Weight change (g) in control (saline) and bleomycin-treated WT and LKO mice during the course of the study. **c** Kaplan–Meier analysis of overall survival in WT and LKO bleomycin-treated mice. **d** Spatial distribution and heatmap of glycogen chain length +7 (1175 *m/z*) in lung tissue from control and bleomycin-treated WT and LKO mice. **e** Glycogen chain length abundance in lung tissue from control and bleomycin-treated WT and LKO mice (*n* = 2 animals for WT-saline, *n* = 3–4 animals/groups for others). Values are presented as mean +/− standard error. **f** Total glycogen abundance in lung tissue from control and bleomycin-treated WT and LKO mice (*n* = 2 animals for WT-saline, *n* = 3–4 animals/groups for others). Values are presented as mean +/− standard error. *p*-value was calculated using one-way ANOVA followed by multiple comparisons test. **g** Representative images of hematoxylin and eosin (H&E) staining of lung tissue from control and bleomycin-treated WT and LKO

mice. **h** Ashcroft scoring for lung tissue from control and bleomycin-treated WT and LKO mice (*n* = 2 animals for WT-saline, *n* = 3–5 animals for other groups). Values are presented as mean +/− standard error. *p*-value was calculated using one-way ANOVA followed by multiple comparisons test. **i** Total collagen from control and bleomycin-treated WT and LKO mice (*n* = 2 animals for WT-saline, *n* = 3–5 animals for other groups). Values are presented as mean +/− standard error. *p*-value was calculated using one-way ANOVA followed by multiple comparisons test. **j** Schematic of MALDI-imaging of collagen peptides. Created with BioRender.com. **k** Spatial distribution and heatmap of collagen peptide 1247 *m/z* and 843 *m/z* in lung tissue from control and bleomycin-treated WT and LKO mice and total abundance. (Putative amino acid sequence, collagen subtype, and # of hydroxylate proline sites (HYP) on are displayed in white text above representative images (*n* = 2 animals for WT-saline, *n* = 3–5 animals for other groups). Values are presented as mean +/− standard error. *p*-value was calculated using one-way ANOVA followed by multiple comparisons test.

shared between PF from different disease origins. These results are particularly exciting, as recent developments of glycogen targeting therapeutics demonstrate efficacy in preclinical models of glycogen storage diseases[84–87], these include multiple modalities that targeting GYS through small molecule inhibition or anti-sense oligonucleotides

with some already being tested in patients[88]. Repurposing these compounds could offer an additional treatment avenue for patients suffering from PF that warrant additional research and development.

In this study, we tested the application of spatial metabolomics for predicting histopathology and biomarker discovery in human

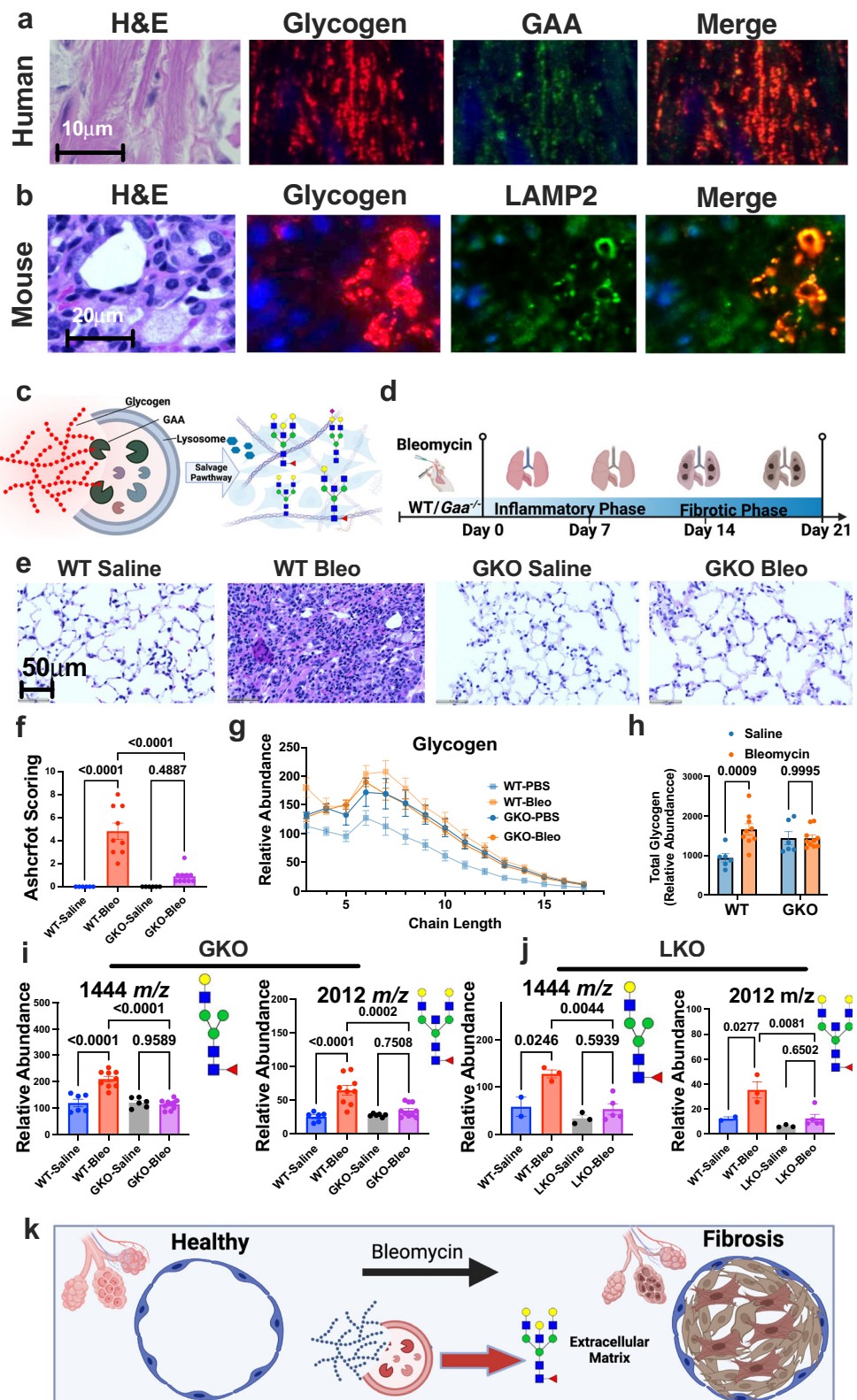

clinical specimens for PF progression. We next demonstrated the biological implication of inhibiting glycogen utilization in the development of PF. One limitation of this study is that the analysis was performed with a step size of 75 μm, above the size of an average human cell of ~5–50 μm[89]. At this visual resolution, we focused on anatomical and histopathological regions on tissue sections. Some newer MALDI-MSI instruments are now capable of sub-cellular imaging (5–20 μm), and future experiments should be performed using these

instruments to reveal single-cell resolution that would aid in the study of immune cell populations within PF. LKO and GAA mouse models are not suitable for aging studies beyond those already included. There are coronary[90], muscular[91], and neuronal[92] issues associated with these mouse models past 8 months of age, therefore inducible models, small molecule inhibitor/activator, or additional in vivo fibrosis models studies should be included for age-related studies in the future. It would also be beneficial to test. Finally, inflammatory cells are critical

**Fig. 6 | Glycogen utilization by lysosomal-GAA during PF in vivo.**
**a** Immunofluorescent/co-localization analysis of glycogen and acid alpha-glucosidase (GAA) from an adjacent section of human IPF lung shown in Fig. 2a. Tissue is stained with glycogen (red), GAA (green), and DAPI (blue).
**b** Immunofluorescent/co-localization analysis of glycogen and LAMP2 from an adjacent section of mouse PF lung. Tissue is stained with glycogen (red), LAMP2 (green), and DAPI (blue). **c** Schematic of lysosomal salvage pathway of glycogen by GAA to provide substrates for other complex carbohydrates. Created with BioRender.com. **d** Schematic of bleomycin-induced lung fibrosis model in wild-type (WT) and Gaa$^{-/-}$ mice (GKO). Created with BioRender.com. **e** Representative images of hematoxylin and eosin (H&E) staining of lung tissue from control (saline) and bleomycin-treated WT and GKO mice. **f** Ashcroft scoring for lung tissue from control and bleomycin-treated WT and LKO mice ($n = 6–10$ animals/group). Values are presented as mean +/− standard error. *p*-value was calculated using one-way ANOVA followed by multiple comparisons test. **g** Glycogen chain length abundance in lung tissue from control and bleomycin-treated WT and LKO mice ($n = 6–10$

animals/group). Values are presented as mean +/− standard error. **h** Total glycogen abundance in lung tissue from control and bleomycin-treated WT and LKO mice ($n = 6–10$ animals/group). Values are presented as mean +/− standard error. *p*-value was calculated using one-way ANOVA followed by multiple comparisons test. **i** Total abundance of 1444 *m/z* and 2012 *m/z* in lung tissue from control and bleomycin-treated WT and GKO mice. Molecular structure of the selected N-linked glycan is to the right of the graph ($n = 6–10$ animals/group). Values are presented as mean +/− standard error. *p*-value was calculated using one-way ANOVA followed by multiple comparisons test. **j** Total abundance of 1444 *m/z* and 2012 *m/z* in lung tissue from control and bleomycin-treated WT and LKO mice. Molecular structure of the selected N-linked glycan is to the right of the graph ($n = 2$ animals for WT-saline, $n = 3–5$ animals for other groups). Values are presented as mean +/− standard error. *p*-value was calculated using one-way ANOVA followed by multiple comparisons test. **k** Schematics of lysosomal salvaging of glycogen as a critical component of PF progression. Created with BioRender.com.

components of the PF progression, while our study focuses on myo-fibroblasts, previous studies have demonstrated the role of glycogen in macrophages and T-cells in other disease settings[93]. Therefore, future directions should include assessing the roles of glycogen in unique immune cell populations driving PF.

## Methods

### Ethics statement
De-identified human patient tissues were obtained from the University of Kentucky Biospecimen Procurement and Translation Pathology Shared Resource Facility operating under the exempt status approved by the intuitional institutional Review Board at University of Kentucky. Due to limited samples available, sex and gender were not considered in the study design.

### Chemicals, reagents, and cell lines
High-performance liquid chromatography-grade acetonitrile, ethanol, methanol, water, trifluoroacetic acid (TFA), bleomycin (B5507, Lot#SLCG3757), and recombinant isoamylase were purchased from Sigma-Aldrich. α-cyano-4-hydroxycinnamic acid (CHCA) matrix was purchased from Cayman Chemical. Histological-grade xylenes were purchased from Spectrum Chemical. Citraconic anhydride for antigen retrieval was obtained from Thermo Fisher Scientific. Recombinant PNGaseF Prime was obtained from N-Zyme Scientifics (Doylestown, PA, USA). Normal lung fibroblast isolated from adult lung tissue and diseased lung fibroblast isolated from adult idiopathic pulmonary fibrosis lung tissue were purchased from Lonza, USA (Cat# CC-2512 and CC-7231, respectively). $^{13}C_6$-glucose was purchased from Cambridge Isotope Laboratories, Inc (Cat# CLM-1396-pk).

### Mouse models
Mice were housed in climate-controlled environment with a 14 (light)/ 10 (dark) hours light/dark cycle with temperature (18–23 °C) and humidity (50–60%) control. Water and solid diet provided ad libitum throughout the study (Tekad #2018). *Epm2a$^{-/-}$* mice, referred to as laforin knockout (LKO), were maintained in house[94]. *Gaa$^{-/-}$* mice were gifts from the late Dr. Peter Roach. Both *Epm2a$^{-/-}$ Gaa$^{-/-}$* animals were generated on the C57BL/6J background. Wild-type (WT) C57BL/6J mice were purchased from Jackson Laboratory. Male and female mice between 3- and 6-months of age were anesthetized by i.p. injection of xylazine (AnaSed, Akorn) and ketamine (Ketathesia, Henry Schein) and received intratracheal (IT) instillation of a single dose of 0.3 U/kg of bleomycin sulfate from Streptomyces verticillus (Sigma-Aldrich) dissolved in 50 μL sterile saline. Control mice received saline IT. Mice were monitored and weighed daily throughout the study. At the experimental endpoint (24 h after 21 days), mice were sacrificed followed by immediate resection of the lungs. Both male and female were used for the study but sex was not considered in the study design as

previous studies suggest no sex differences in this mouse model of fibrosis at the age we are using[95]. The University of Kentucky Institutional Animal Care and Use Committee has approved all of the animal procedures under the protocol number 2017-2792.

### Tissue procurement
Mice were sedated with ketamine (100 mg/kg) and xylazine (10 mg/kg) followed by exsanguination and immediate resection of the lungs which were fixed in neutral-buffered formalin (NBF) for 24 h then switched to 70% ethanol and paraffin-embedded for long-term storage. De-identified human patient tissues were obtained from the University of Kentucky Biospecimen Procurement and Translation Pathology Shared Resource Facility. Human tissues were also preserved in NBF and paraffin-embedded for long-term storage. All hematoxylin & eosin (H&E) staining was performed by the University of Kentucky Biospecimen Procurement and Translation Pathology Shared Resource Facility using the method previously described[96].

### Formalin-fixed paraffin-embedded slide preparation for MALDI-MSI
Tissues were sectioned at 4 μm and mounted on positively charged glass slides for MALDI imaging as previously described[24]. Slides were heated at 60 °C for 1 h. After cooling, tissue sections were deparaffinized by washing twice in xylene (3 min each). Tissue sections were then rehydrated by washing slides twice in 100% ethanol (1 min each), once in 95% ethanol (1 min), once in 70% ethanol (1 min), and twice in water (3 min each). Following washes, slides were transferred to a coplin jar containing citraconic anhydride buffer for antigen retrieval and the jar was placed in a vegetable steamer for 25 min. Citraconic anhydride buffer was prepared by adding 25 μL citraconic anhydride in 50 mL water and adjusted to pH 3.0 with HCl. After antigen retrieval, slides were dried in a vacuum desiccator prior to enzymatic digestion.

### Glycogen and N-glycan MALDI-mass spectrometry imaging
An HTX spray station (HTX) was used to coat the slide with a 0.2 ml aqueous solution of isoamylase (3 units/slide) and PNGase F (20 mg total/ slide). The spray nozzle was heated to 45 °C with a spray velocity of 900 m/min. Following enzyme application, slides were incubated at 37 °C for 2 h in a humidified chamber, and dried in a vacuum desiccator prior to matrix application [α-cyano-4-hydroxycinnamic acid matrix (0.021 g CHCA in 3 ml 50% acetonitrile/50% water and 12 μL 25% TFA) applied with HTX sprayer]. For detection and separation of glycogen and N-glycans, a Waters SynaptG2-Si high-definition mass spectrometer equipped with traveling wave ion mobility was used. The laser was operating at 1000 Hz with an energy of 200 AU and spot size of 75 μm, mass range is set at 500–3000 *m/z*. Ion mobility setting were done according to previously established parameters[17,24] with a trap entrance energy of 2 V, trap bias of 85 V, and DC/exist of 0 V. Wave

velocity settings were set to: trap 9.6 m/s, IMS 4.6 m/s, transfer 17.4 m/s. Wave height settings were set to: trap 4 V, IMS, 42.7, transfer 4 V, additional settings are variable wave ramp down of 1400 m/s. Data was acquired with Masslynx v4.2. MALDI images were produced using the HDI software v1.5 (Waters Corp) following built in peak integration function to account for mass drift over the MALDI run. All MALDI images were normalized to total ion current (TIC) within each pixel.

## Quantitative glycogen MALDI-mass spectrometry imaging

1, 10, 20, 40, 100, 1000 ng of purified rabbit liver glycogen were directly spotted on the microslide adjacent to the tissue section. An HTX spray station (HTX) was used to coat the slide with a 0.2 ml aqueous solution of isoamylase (3 units/slide). The spray nozzle was heated to 45 °C with a spray velocity of 900 m/min. Following enzyme application, slides were incubated at 37 °C for 2 h in a humidified chamber, and dried in a vacuum desiccator prior to matrix application [α-cyano-4-hydroxycinnamic acid matrix (0.021 g CHCA in 3 ml 50% acetonitrile/50% water and 12 μL 25%TFA) applied with HTX sprayer]. For detection and separation of glycogen, Bruker timeTOF Flex mass spectrometer was used. The laser was operating at 10,000 Hz and 300 shots/second, mass range is set at 700–3000 $m/z$. Data was acquired with timsControl v4.1.12 and Fleximaging v7.2. MALDI images were produced using the SCiLS v2023a. Pixel information was exported using the SCILS API package in R studio and analysis was performed in prism. The molecular ion of 1175 $m/z$ that corresponds to glucose polymer 6 was used to produce standard curve and absolute quantitation of glycogen levels in situ.

## MALDI-MSI dataset processing as part of user input module

MALDI-MSI data files were processed to adjust for mass drift during the MALDI scan to enhance image quality and improve signal-to-noise ratio that will assist in the machine learning module. Raw MALDI-MSI data files were processed using an algorithm available within the HDI software (Waters Corp). To adjust for mass drift during the MALDI scan, raw files were processed using a carefully curated and established list of 50 MALDI matrix peaks ($m/z$) and 155 glycogen and N-linked glycan peaks ($m/z$) listed in Supplemental Table S1. Files were processed at a sample duration of 10 sec at a frequency rate of 0.5 min, and an $m/z$ window of <1 Da, using an internal lock mass of previously defined N-linked glycan 1257.4296 $m/z$ with a tolerance of 1 amu and a minimum signal intensity of 100,000 counts. Three to five glycans were spot checked for abundance and known spatial distribution as a positive control[24]. Post-processed MALDI data files were exported in a tabular format as .CSV for input into the machine learning module.

## Dimensionality reduction and clustering as part of computer-based module

Normalization, high-dimensional clustering, and UMAP and spatial plots, were done using the Python® software environment[97]. First, each complex carbohydrate feature was normalized on total ion count (TIC) within individual pixels. The Leiden algorithm[36] automatically performed spatially agnostic clustering of pixels and optimized cluster approximation based on similarities in their carbohydrate features. Uniform Manifold Approximation and Projection (UMAP)[37] was applied to embed high-dimensional carbohydrate features of pixels in a low dimensional space for data visualization and interpretation. The clustering results were visualized by a pixel-colored UMAP plot in 2D feature space and a pixel-colored spatial plot mapping pixel to the original X-Y coordinates recorded by MALDI-MSI. Both plots are qualified for revealing tissue anatomy, histopathology, and heterogeneity. Leiden clustering algorithm and UMAP plot generation are built-in modules as part of the Scanpy package available within Python®[98]. Code is available for free at github.com/maldiUKY/HDC-SC. Codes are tested and functional under the MacBook Pro IOS environment.

## Discovery of overrepresented complex carbohydrate features and pathway enrichment

Multivariate analyses of all complex carbohydrate features within unique clusters were analyzed using the Metaboanalyst 5.0 software ($n = 3$ technical regions of interest [ROIs] per cluster or biological sample) as previously described[99]. Log transformation and auto scaling were used for normalization. Heatmaps were generated with the top 50 ranked features and hierarchical clustering was performed using the Euclidean distance measure and Ward linkage. Pathway enrichment analysis based on overrepresented molecular features were done manually based on previously established pathways of N-linked glycosylation and glycogen metabolism[100]. All MALDI-MSI images of glycogen and N-glycans were generated using the HDI software (Waters Corporation). Representative glycan structures were generated in GlycoWorkbench.

## Histopathology assessments

Histopathology assessments were performed by four board-certified clinical pathologists independently to improve rigor and reproducibility. All H&E slides were assessed in a CLIA certified pathology laboratory using an inverted brightfield microscope (Olympus BX43). Image analysis was performed using the HALO image analysis software (Indica labs). Assessments were made based on the spatial clustering produced by HDR-SC and histopathology assessment from an immediate adjacent section 4μm apart.

## Immunohistochemistry

Fixed lung tissues were sectioned at 10 μm and immunohistochemistry was performed at the Biospecimen Procurement and Translation Pathology Shared Resource Facility using the method previously described[17]. Briefly, tissue was rinsed with 0.01 M PBS, incubated for 1 h in 5% normal goat serum in 0.3% Triton X-100 in PBS. The tissue was then incubated at room temperature in the primary antibody (below) at diluted in 1% normal goat serum, followed by 3 × 15 min rinses in 0.05% Tween-20 in PBS. Tissue was then incubated for 1 h in secondary antibody diluted in 1% normal goat serum in PBS. Antibodies used for other markers are glycogen synthase (LSBio, Cat# LS-B12901), glycogen phosphorylase brain isoform (LSBio, Cat# LS-B4749), and glycogen phosphorylase muscle isoform (Protech, 1Cat# 9716-AP). Digital images were acquired through the ZEISS Axio Scan.Z1 high-resolution slide scanner. Quantitative image analysis was performed using the Halo software (Indica labs) using the multiplex IHC modules.

## Co-Immunofluorescence

Fixed brains were embedded in paraffin and sectioned at 4 μm onto slides by the University of Kentucky Biospecimen Procurement and Translation Pathology Shared Resource Facility. Slides were dewaxed, rehydrated, and heated to 60 °C for 30 min in citraconic anhydride buffer (pH 6.5) for antigen retrieval. After cooling, slides were incubated in primary antibody followed by incubation with a secondary antibody. Glycogen (clone: IV58b6, dilution: 1:500), GAA (Cat#Proteintech 14267-1-AP, dilution: 1:100), LAMP2 (Cat# Abcam ab125068, dilution: 1:100), Alpha smooth muscle actin (Cat# GTX100034, dilution: 1:100), DAPI (Cat# Novus NBP2-31156, dilution: 1:5000), Goat anti-rabbit Alexa 488 (Cat# A-11034, dilution: 1:400), Goat anti-mouse Alexa 647 (Cat# A-21236, dilution: 1:400). Slides were then cover slipped using Southern Biotech DAPI Fluoromount-G (Cat. 591 #0100-20). Digital images were acquired as 12 Z-stacks through the Zeiss Axio Scan Z.7 digital slide scanner at 40X magnification and processed using HALO software (v3.3.2541.345, Indica Labs, Albuquerque, NM).

## Lung fibrosis assessments

Lung fibrosis was evaluated by Ashcroft score using H&E sections. For analysis, two images from four micrographs of whole lung sections were randomly selected from four mice. These randomly selected

images were individually assessed in a blinded manner with an index of 0–8: 0, normal lung; 1, minimal fibrous thickening of alveolar or bronchiolar walls; 2–3, moderate thickening of walls without obvious damage to lung architecture; 4–5, increased fibrosis with definite damage to lung structure and formation of fibrous bands or small fibrous masses; 6–7, severe distortion of structure and large fibrous areas, evidence of "honeycomb lung"; 8, total fibrous obliteration of the field. Soluble collagen content from bronchoalveolar lavage fluid (BALF) was determined using the Sircol assay (Biocolor), according to the manufacturer's instructions. Sircol dye bound to collagen was evaluated by a microplate reader at 555 nm. BALF was collected by performing two washes of 0.7 mL with PBS + EDTA (0.1 mM) on ice.

## Collagen targeted imaging proteomics
Samples were prepared as previously described for targeted collagen and extracellular matrix peptide imaging[55,101–104]. Briefly, deglycosylated samples[105,106] were antigen retrieved with 10 mM Tris, pH 9, autosprayed (M5, HTX-Technologies) with 0.1 µg/µL collagenase type III (Worthington) dissolved in ammonium bicarbonate pH 7.4, 1 mM $CaCl_2$, and incubated at 38.5 °C with ≥85% humidity for 5 h. The matrix α-Cyano-4-hydroxycinnamic acid (CHCA, Sigma-Aldrich) was dissolved in 1.0% trifluoracetic acid (Sigma), 50% acetonitrile (LC-MS grade, Fisher Chemical) and autosprayed (M5, HTX Technologies) onto tissue. Tissues were imaged on a MALDI-QTOF (timsTOF-flex, Bruker) in positive ion mode over $m/z$ range 700–2500. Laser was adjusted to 20 µm² and each pixel consisted of 300 laser shots. Images were collected with a laser step size of 60 µm. Data was visualized in SCiLS Lab Software (v2022b, Bruker) and processed for image segmentation and principal component analysis. Peak data were exported by mean spectrum processed as peak maximum and further statistical comparisons were done using Metaboanalyst 5.0 and GraphPad Prism 9.0. Exported peak intensities were visualized as heatmaps using the TM4 MultiExperiment Viewer suite[107] with clustering by Manhattan metric and single linkage.

## Cell culture and 13C-glycogen tracing in fibroblast cell lines
Normal and diseased lung fibroblast were maintained in fibroblast growth medium (Lonza, Cat# CC-3132) supplement with the FGM-2 SingleQuots supplement (Lonza, Cat# CC-4126) in 10 cm dishes. For the $^{13}$C-glycogen enrichment experiment, fibroblast cells were allowed to reach ~50% confluency, followed by the addition of DMEM base media supplemented with 10 mM $^{13}C_6$-glucose, 2 mM Gln, 10% dialyzed fetal bovine serum in a $CO_2$ incubator maintained at 37 °C. At the designated timepoints, cells were washed with cold PBS three times followed by extraction with 50% methanol and separated into polar, insoluble fraction that contains the glycogen. For the $^{13}$C-glycogen washout experiment, fibroblast cells were grown in $^{13}C_6$-glucose enrichment media for 48 h followed by the replacement of DMEM base media supplement with $^{12}$C-glucose. At designated timepoints, cells were washed with cold PBS three times followed by extraction with 50% methanol and separated into polar and the glycogen-containing insoluble fraction.

## Gas-chromatography-mass spectrometry (GCMS) analysis of 13C-enriched glycogen
GCMS assessment of glycogen was done using method previously described[56,57]. Briefly, insoluble fraction was subject to acid hydrolysis with 2N HCL for 3 h at 98 °C. The reaction was quenched with 2N NaOH for subsequent experiments. For GC-MS analysis, samples were first dried in a SpeedVac (Thermo), followed by sequential addition of 20 mg/ml methoxyamine hydrochloride in pyridine, and then the trimethylsilylating agent N-methyl-N- trimethylsilyl-trifluoroacetamide (MSTFA) was added followed by GCMS analysis. GC-MS protocols were similar to those described previously[56,57]. The electron ionization (EI) energy was set to 70 eV. Scan ($m/z$: 50–800) and selected ion

monitoring mode were used for qualitative measurement and isotope monitoring, respectively. Ions used for the enrichment of glycogen-derived glucose are 319 (unlabeled) and 320, 321, 322, 323, (labeled). Batch data processing and natural $^{13}$C labeling correction were performed using the Data Extraction for Stable Isotope-labeled metabolites (DExSI) software package.

## Targeted analysis of metabolites by liquid-chromatography-mass spectrometry
Approximately 20 mg of pulverized, frozen tissue was extracted in 3 ml ice-cold 60% acetonitrile by extensively vortexing. Next, 1 ml chloroform was added to each sample and samples were mixed by manual shaking followed by centrifuging at $3200 \times g$ for 20 min at 4 °C. The aqueous layer was moved to a new tube and lyophilized. The middle layer containing protein and insoluble material was transferred to a new tube, washed twice with 50% methanol, once with 100% methanol, and briefly dried in a speed-vac. After drying, the insoluble material was hydrolyzed by heating samples in 3N HCl at 95 °C for 2 h. 100% methanol was added to hydrolysate to achieve a final concentration of 50% methanol, samples mixed, centrifuged at $18,000 \times g$ for 10 min at 4 °C, then supernatant dried on a speed vac. Polar metabolites were separated by HILIC chromatography using an Agilent InfinityLab Poroshell 120 HILIC-Z column (2.7 m, 2.1 × 150 mm) with a binary solvent system of 10 mM ammonium acetate in water, pH 9.8 (solvent A) and ACN (solvent B) with a constant flow rate of 0.25 ml/min. The column was equilibrated with 90% solvent B. Polar metabolites were resuspended in a 1:1 mixture of solvents A:B, centrifuged at $18,000 \times g$ for 5 min at 4 °C, then transferred to glass vials. 4 µl of each sample was injected and the gradient proceeded from 0–15 min linear ramp 90%B to 30%B, 15–18 min isocratic flow of 30%B, 18–19 min linear ramp from 30%B to 90%B, and 19–27 min column regeneration with isocratic flow of 90%B. Metabolites were measured using an Agilent 6545 quadrapole-time of flight mass spectrometer (MS) coupled to an Agilent 1290 Infinity II UHPLC. Individual samples and standards were acquired with full scan MS in negative mode. Peaks for the deprotonated [M-H]⁻ ions were extracted and integrated using Agilent Qualitative Analysis software V10.0 (retention time, formula, and $m/z$ are presented in Supplementary Table 2). Polar metabolite peaks were normalized to the amino acid content in the hydrolyzed insoluble biomass fraction. Samples were resuspended in 70 µl of 20 mg/ml methoxyamine in pyridine and incubated for 1.5 h at 37 °C. Samples were centrifuged at $18,000 \times g$ for 5 min and 50 µl of each sample was transferred to a glass vial. 80 µl of N-Methyl-N-(trimethylsilyl)trifluoroacetamide + 1% chlorotrimethylsilane (Themo Scientific) was added to each sample and incubated at 37 °C for 30 min. Samples were then analyzed by gas chromatography mass spectrometry (GCMS) for as described above.

## Statistical analysis
All dimensionality reduction and clustering analyses were performed using the Python® software environment. Both clustering and differential analyses were conducted with the Scanpy package by Leiden algorithm and Wilcoxon rank-sum test, respectively. For Metaboanalyst multivariate analyses of all glycans, log transformation, and auto scaling were used for normalization. Heatmaps were generated based on the Euclidean distance measure and the Ward clustering algorithm. Glycans with variable importance in projection (VIP) scores >1.5 based on partial least squares discriminant analysis (PLS-DA) were selected for further analysis. For biomarker analysis, areas under the curve (AUC) were obtained using multivariate receiver operating characteristic (ROC) analysis based on the linear SVM classification method and SVM feature ranking method. Statistical analysis for discovery of overrepresented complex carbohydrate features in patient samples and in vivo studies were carried out using GraphPad Prism 9.0. Sample sizes were chosen based on previously published results. Blinding is

incorporated in all experiments; no data were excluded from the analyses. All numerical data are represented as mean ± S.E.M. Column analysis was performed using ANOVA or t-test. The statistical parameters for each experiment can be found in the figures and figure legends. All cell line experiments were repeated independently for three times, all animal experiments were repeated independently twice with similar results.

### Reporting summary
Further information on research design is available in the Nature Portfolio Reporting Summary linked to this article.

## Data availability
The MALDI imaging data generated in this study have been deposited in the DRYAD database under[108] https://doi.org/10.5061/dryad.jwstqjqfg. MALDI imaging data exported as .txt format are provided in the Source data file. The pool metabolomics data used in this study are available in the Metabolomics Workbench database under accession code/study ID ST002547 [https://doi.org/10.21228/M8QF0W]. Source data are provided with this paper.

## Code availability
Python code for high-dimensional clustering and spatial clustering analysis of MALDI imaging is deposited and available at github.com/maldiUKY/HDC-SC. Example datasets in .txt format can be found in the source file.

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

## Acknowledgements

We would like to thank Dr. Craig Vander Kooi for vigorous discussions regarding the work, Mrs. Dana Napier for performing histological staining on tissue slices, and the Markey Cancer Center. This study was supported by National Institute of Health (NIH) grants R01AG066653, R01CA266004, R01AG078702, CureAlz fund, St Baldrick's Career Development Award, V-Scholar Grant, Rally Foundation Independent Investigator Grant to R.C.S., HL131526 and HL151419 to C.M.W., R21NS121966 to W.J.A., R35NS116824 to M.S.G., L.E.A.Y. was supported by NIH/NCI F99CA264165, L.R.C. was supported by NIH/NCI training grant T32CA165990, and J.B.D. was supported by NIH training grant T32GM132055. This research was also supported by funding from the University of Kentucky Markey Cancer Center and the NIH-funded Biospecimen Procurement & Translational Pathology Shared Resource Facility of the University of Kentucky Markey Cancer Center P30CA177558. S.S.V. was granted with a sabbatical from Federal University of Rio de Janeiro, Brazil.

## Author contributions

Conceptualization, R.C.S; methodology, R.C.S., J.L., and C.M.W.; investigation, L.R.C., D.B.A., Q.S., S.S.V., B.E.D., T.R.H., R.C.S., L.E.A.Y., J.E.F., A.V.H., J.B.D.; R.J.M., K.J.A., K.H.M., J.A.J., W.J.A., P.M.A.; C.M.W., M.S.G., J.B.D., P.M.A., R.C.B., and H.A.C.; writing—original draft, R.C.S.; writing—review & editing, R.C.S., L.R.C., Q.S., W.J.A., C.M.W., M.S.G., and D.B.A.; funding acquisition, R.C.S.; resources, R.C.S.; supervision, R.C.S.

## Competing interests

R.C.S. has research support and received consultancy fees from Maze Therapeutics. D.B.A. received book royalty from Wolters Kluwer. R.C.S., M.S.G., and R.C.B. are co-founders of Attrogen LLC. R.C.S. is a member of the Medical Advisory Board for Little Warrior Foundation. W.J.A. has a sponsored research agreement with BioMarin Pharmaceutical Inc. for a project unrelated to the material addressed in the present manuscript. M.S.G. has research support and research compounds from Maze Therapeutics, Valerion Therapeutics, Ionis Pharmaceuticals. M.S.G. also received consultancy fee from Maze Therapeutics, PTC Therapeutics, and the Glut1-Deficiency Syndrome Foundation. The remaining authors declare no competing interests.
