## [Peer Review File · Nature Communications]

Spatial metabolomics reveals glycogen as an actionable target for pulmonary fibrosisEditorial Note: This manuscript has been previously reviewed at another journal that is not operating a transparent peer review scheme. This document only contains reviewer comments and rebuttal letters for versions considered at *Nature Communications* .

REVIEWER COMMENTS

Reviewer #1 (Remarks to the Author):

The authors have addressed all my questions.

Reviewer #4 (Remarks to the Author):

In the manuscript by Conroy et al, the authors developed a novel tool to study pulmonary fibrosis progression and identified glycogen as a potential target. The authors showed that glycogen is increased in fibrotic lungs in both human and mice with a novel MALDI-MSI method. However, the biological significance of this observation is not fully elucidated. The detailed comments are as follows:

1. The authors need to show the absolute amount of glycogen level in normal/patient samples, and in WT and KO mice. The difference in the glycogen amount will give a biological sense of whether it is a critical driver of fibrosis as claimed, or merely a byproduct of this process. The glycogen staining method is based on an antibody that is only verified from limited publications. It is questionable how reliable this antibody is, and must be validated here.

2. In the patient samples, glycogen accumulated exclusively in myofibroblasts (Fig 2O). However, this is not the case in mice (Fig 4H). How is this difference explained? Is the increase of glycogen only a readout of fibroblast number in fibrosis?

3. The author claimed that the turnover of glycogen is the key to the progression of fibrosis. However, this conclusion is an overstatement for two reasons. First, no evidence is presented to support that glycogen is correlated with the severity of fibrosis or disease progression at different stages. More time points are needed. Second, there is no data showing that the metabolic product of glycogen turnover is changed in fibrosis in humans or mice.

4. In Fig 3C and 4F, the authors concluded that "significantly increased glycogen chain length distribution in fibrotic cores compared to normal lung tissues". The reviewer finds that the distribution is in fact very similar in between the two groups in both figures. These data should be presented in histogram form with percentage instead of dot plots of abundance. If correct plot is used, it is easier to see that the difference between groups is coming from the absolute amount of glycogen, not distribution.

5. The authors examined PYGB level in the lung and found no difference in normal and diseases tissues. However, PYGB is not highly expressed in the lung. Instead, PYGM expression is higher in the lung, especially in fibroblasts.

6. Diseased tissues often have higher metabolic rates. To demonstrate that glycogen metabolism is specifically upregulated in lung fibrosis and therefore a "therapeutic target", it would be crucial to show at least one other metabolite not affected in the fibrotic lungs as a control.

Reviewer #5 (Remarks to the Author):

In this manuscript Conroy et al., demonstrated the application of high-dimensionality reduction and spatial clustering (HDR-SC) and histopathological annotation/prediction of MALDI-MSI datasets to

assess tissue metabolic heterogeneity in human lung diseases. They showed that glycogen and N-linked glycans is a critical metabolic process favoring pulmonary fibrosis progression. They used two different knock out model to substantiate their claim. It's a new technique used to study the progression of lung fibrosis in paraffin embedded or frozen tissue section, which has clinical significance, but overall, the significance and innovation of this present study is minimal. Elevation of glycogen level in IPF is already known and targeting Glycogen Synthase Kinase (GSK) ameliorates pulmonary fibrosis in mice already reported (Scientific Report 2019 Dec 12;9(1):18925. doi: 10.1038/s41598-019-55176-w. PMID: 31831767). Here the authors only measured glycogen and N-linked glycans in different experimental set up which is not sufficient. They need more depth experimental set up to elucidate the role of glycogen metabolism in pulmonary fibrosis, their mechanism of action. As bleomycin model of pulmonary fibrosis in young mice doesn't depict the IPF. They need to consider aged mice in their experimental set up. Overall, the significance of the study is low with minimal new information generated. Thus, in its current form and the experimental set up, it's not suitable for publication in nature communication.

We would like to thank the reviewers for his/her enthusiasm regarding our manuscript. We have carefully considered each comment and addressed all points with additional data to improve the overall quality of the manuscript. Please find the reviewer's comments are in grey, and we respond to reviewer's comments in black.

Reviewer #4 (Remarks to the Author):

In the manuscript by Conroy et al, the authors developed a novel tool to study pulmonary fibrosis progression and identified glycogen as a potential target. The authors showed that glycogen is increased in fibrotic lungs in both human and mice with a novel MALDI-MSI method. However, the biological significance of this observation is not fully elucidated. The detailed comments are as follows:

1. The authors need to show the absolute amount of glycogen level in normal/patient samples, and in WT and KO mice. The difference in the glycogen amount will give a biological sense of whether it is a critical driver of fibrosis as claimed, or merely a byproduct of this process. The glycogen staining method is based on an antibody that is only verified from limited publications. It is questionable how reliable this antibody is, and must be validated here. We appreciate the reviewer's expert feedback regarding absolute amount of glycogen levels in human and mice. To address this, we developed the quantitative glycogen MALDI imaging assay where increasing amounts of purified glycogen are directly spotted adjacent to tissue sections on the same slide followed by isoamylase treatment and MALDI imaging. Using this method, we established the dynamic linear range for glycogen from 1ng to 1000ng. Further, we demonstrated the absolute amount glycogen of ~500ng/pixel in fibrotic regions, and ~90ng/pixel in non-fibrotic regions in human specimens. These data are included as supplementary figure 2N and presented below.

Quantitative glycogen MALDI imaging in situ. Increasing amounts of purified glycogen were spotted adjacent to the tissue section and used to generate the standard curve. Absolute glycogen levels in fibrosis and non-fibrotic regions were determined using the equation derived from the line of best fit.

2. In the patient samples, glycogen accumulated exclusively in myofibroblasts (Fig 2O). However, this is not the case in mice (Fig 4H). How is this difference explained? Is the increase of glycogen only a readout of fibroblast number in fibrosis?

We would like to thank the reviewer for raising this point. We have previously demonstrated that the normal lung cells contain detectable glycogen by antibody¹ and MALDI imaging². We also measured basal glycogen amount to be around 200ng/pixel in the normal mouse lung while fibrotic regions contain almost ~600ng/pixel. We are actively working on the roles of glycogen in the developmental stages of the mouse lung and will report these data in future manuscripts. Glycogen presence in the non-fibrotic regions of human specimens is below the detection limit of the antibody.

3. The author claimed that the turnover of glycogen is the key to the progression of fibrosis. However, this conclusion is an overstatement for two reasons. First, no evidence is presented to support that glycogen is correlated with the severity of fibrosis or disease progression at different stages. More time points are needed. Second, there is no data showing that the metabolic product of glycogen turnover is changed in fibrosis in humans or mice.

I hope the reviewer can appreciate that defining glycogen in different fibrosis stages is a challenging task. 1) Human samples are collected postmortem or at lung transplant where most disease are end stage. 2) there are no mouse models of late-stage fibrosis. However, working with our clinical pathologist, Dr. Derek Allison, we identified early, mid and end stage fibrotic regions within the human specimens. We demonstrated step wise decrease in glycogen from early to end stage fibrosis, further support that glycogen is a critical part of fibrosis progression. Finally, we performed targeted LCMS analysis using PBS and bleomycin treated mice and showed significant reduction in metabolic products of glycogen

turnover such as UDP-glucose and glucose-6-phosphate. These data are incorporated as supplementary figure 4F and 6G-H respectively, and are presented below.

Top: Hematoxylin and eosin (H&E) staining showing early, mid, and end stage fibrosis in human PF specimens. **Bottom:** Representative images and relative abundance of glycogen chain length +7 (1175 m/z) and a biantennary glycan spatial distribution in in early-, mid-, and end-stage fibrosis. Molecular structure of the selected N-linked glycan is to the right of the heatmap.

Targeted liquid-chromatography mass spectrometry analysis of polar metabolites associated with glycogen metabolism from PBS and bleomycin treated mice. UDP-glucose, glucose-6-phosphate, and fructose-6-phosphate are shown.

4. In Fig 3C and 4F, the authors concluded that “significantly increased glycogen **chain length distribution** in fibrotic cores compared to normal lung tissues”. The reviewer finds that the distribution is in fact very similar in between the two groups in both figures. These data should be presented in histogram form with percentage instead of dot plots of abundance. If correct plot is used, it is easier to see that the difference between groups is coming from the absolute amount of glycogen, not distribution.

We apologize for the miss use of word distribution, we have reworded in our text to indicate an increase in the abundance of different glycogen chain lengths. See example below.

“In addition, glycogen structure analysis shown significantly increases across the spectrum of glycogen chain lengths in fibrotic cores compared to normal lung tissue (Fig. 3C).”

5. The authors examined PYGB level in the lung and found no difference in normal and diseases tissues. However, PYGB is not highly expressed in the lung. Instead, PYGM expression is higher in the lung, especially in fibroblasts.

We appreciate this insight from the reviewer and performed IHC analysis of PYGM. These data are incorporated in supplementary figure 6A and B and are presented below.

Immunohistochemical staining and intensity analyses of glycogen synthase (GYS1), glycogen phosphorylate brain isoform (PYGB), glycogen phosphorylate muscle isoform (PYGM) in fibrotic regions and adjacent alveoli structures.

6. Diseased tissues often have higher metabolic rates. To demonstrate that glycogen metabolism is specifically upregulated in lung fibrosis and therefore a “therapeutic target”, it would be crucial to show at least one other metabolite not affected in the fibrotic lungs as a control.

We appreciate the reviewer’s comment. In the Targeted LCMS analysis, we saw no change in the total pool of alanine, glutamic acid, and glutamine. They are incorporated as supplementary figure 4F are presented below.

Targeted liquid-chromatography mass spectrometry analysis of polar metabolites from PBS and bleomycin treated mice. Amino acids such as alanine, glutamic acid, and glutamine are shown.

Reviewer #5 (Remarks to the Author):

In this manuscript Conroy et al., demonstrated the application of high-dimensionality reduction and spatial clustering (HDR-SC) and histopathological annotation/prediction of MALDI-MSI datasets to assess tissue metabolic heterogeneity in human lung diseases. They showed that glycogen and N-linked glycans is a critical metabolic process favoring pulmonary fibrosis progression. They used two different knock out model to substantiate their claim. It’s a new technique used to study the progression of lung fibrosis in paraffin embedded or frozen tissue section, which has clinical significance, but overall, the significance and innovation of this present study is minimal. Elevation of glycogen level in IPF is already known and targeting Glycogen Synthase Kinase (GSK) ameliorates pulmonary fibrosis in mice already reported (Scientific Report 2019 Dec 12;9(1):18925. doi: 10.1038/s41598-019-55176-w. PMID: 31831767). Here the authors only measured glycogen and N-linked glycans in different experimental set up which is not sufficient. They need more depth experimental set up to elucidate the role of glycogen metabolism in pulmonary fibrosis, their mechanism of action. As bleomycin model of pulmonary fibrosis in young mice doesn’t depict the IPF. They need to consider aged mice in their experimental set up. Overall, the significance of the study is low with minimal new information generated. Thus, in its current form and the experimental set up, it’s not suitable for publication in nature communication.

We would like to politely rebut reviewer #5’s comments.

We believe there is some confusion regarding the literature surrounding glycogen as a metabolite and glycogen synthase kinase (GSK) as a signaling enzyme within in pulmonary fibrosis. GSK is a well-known upstream kinase that regulates

CREB, Tau, Akt through phosphorylation and intertwined with Akt cascade pathway as well³⁻⁷. In the paper that reviewer #5 highlighted, they speculate that “GSK acts through the CREB signaling in pulmonary fibrosis”. In fact, the metabolite glycogen was not mentioned at all in the results or discussion of the manuscript.

Bleomycin has been used as a pulmonary fibrosis model in the community for many years with many high impact and highly cited publications⁸⁻¹¹. We carefully placed our findings in the context of pulmonary fibrosis throughout the manuscript and not idiopathic pulmonary fibrosis (IPF), which is more of an aging related disease.

Finally, LKO and GAA mouse models are not suitable for aging studies beyond those already included. There are coronary¹², muscular¹³, and neuronal¹⁴ issues associated with these mouse models past 8 months of age that makes them unsuitable for an aging study. We believe both mouse models within the manuscript provide strong support for the glycogen utilization through lysosomal pathway as a critical metabolic pathway in pulmonary fibrosis.

References:

- 1 Young, L. E. *et al.* In situ mass spectrometry imaging reveals heterogeneous glycogen stores in human normal and cancerous tissues. *EMBO Molecular Medicine* **14**, e16029 (2022).
- 2 Sun, R. C. *et al.* Nuclear glycogenolysis modulates histone acetylation in human non-small cell lung cancers. *Cell metabolism* **30**, 903-916. e907 (2019).
- 3 Doble, B. W. & Woodgett, J. R. GSK-3: tricks of the trade for a multi-tasking kinase. *Journal of cell science* **116**, 1175-1186 (2003).
- 4 Harwood, A. J. Regulation of GSK-3: a cellular multiprocessor. *Cell* **105**, 821-824 (2001).
- 5 Woodgett, J. R. Judging a protein by more than its name: GSK-3. *Science's STKE* **2001**, re12-re12 (2001).
- 6 Zhou, B. P. *et al.* Dual regulation of Snail by GSK-3 β -mediated phosphorylation in control of epithelial–mesenchymal transition. *Nature cell biology* **6**, 931-940 (2004).
- 7 Kaidanovich-Beilin, O. & Woodgett, J. R. GSK-3: functional insights from cell biology and animal models. *Frontiers in molecular neuroscience* **4**, 40 (2011).
- 8 Izbicki, G., Segel, M., Christensen, T., Conner, M. & Breuer, R. Time course of bleomycin-induced lung fibrosis. *International journal of experimental pathology* **83**, 111-119 (2002).
- 9 Mouratis, M. A. & Aidinis, V. Modeling pulmonary fibrosis with bleomycin. *Current opinion in pulmonary medicine* **17**, 355-361 (2011).
- 10 Moeller, A., Ask, K., Warburton, D., Gauldie, J. & Kolb, M. The bleomycin animal model: a useful tool to investigate treatment options for idiopathic pulmonary fibrosis? *The international journal of biochemistry & cell biology* **40**, 362-382 (2008).
- 11 Adamson, I. Y. & Bowden, D. H. The pathogenesis of bleomycin-induced pulmonary fibrosis in mice. *The American journal of pathology* **77**, 185 (1974).
- 12 Bijvoet, A. G. *et al.* Generalized glycogen storage and cardiomegaly in a knockout mouse model of Pompe disease. *Human molecular genetics* **7**, 53-62 (1998).
- 13 Zhu, Y. *et al.* Glycoengineered acid α -glucosidase with improved efficacy at correcting the metabolic aberrations and motor function deficits in a mouse model of Pompe disease. *Molecular Therapy* **17**, 954-963 (2009).
- 14 Lee, N.-C. *et al.* A neuron-specific gene therapy relieves motor deficits in Pompe disease mice. *Molecular neurobiology* **55**, 5299-5309 (2018).